# S$^2$AC: Energy-Based Reinforcement Learning with Stein Soft Actor Critic

**Safa Messaoud**[1†], **Billel Mokeddem**[1*], **Zhenghai Xue**[2*], **Linsey Pang**[3], **Bo An**[4,2],

**Haipeng Chen**[5†], **Sanjay Chawla**[1†]

[1]Qatar Computing Research Institute, Hamad Bin Khalifa University, [2]School of Computer Science and Engineering, Nanyang Technological University, [3]SalesForce, [4]Skywork AI, [5]Data Science, William & Mary
{smessaoud,bmokeddem,schawla}@hbku.edu.qa, zhenghai001@e.ntu.edu.sg
panglinsey@gmail.com, boan@ntu.edu.sg, hchen23@wm.edu
* Equal contribution   † Corresponding authors

## Abstract

Learning expressive stochastic policies instead of deterministic ones has been proposed to achieve better stability, sample complexity, and robustness. Notably, in Maximum Entropy Reinforcement Learning (MaxEnt RL), the policy is modeled as an expressive Energy-Based Model (EBM) over the Q-values. However, this formulation requires the estimation of the entropy of such EBMs, which is an open problem. To address this, previous MaxEnt RL methods either implicitly estimate the entropy, resulting in high computational complexity and variance (SQL), or follow a variational inference procedure that fits simplified actor distributions (*e.g.*, Gaussian) for tractability (SAC). We propose Stein Soft Actor-Critic (S$^2$AC), a MaxEnt RL algorithm that learns expressive policies without compromising efficiency. Specifically, S$^2$AC uses parameterized Stein Variational Gradient Descent (SVGD) as the underlying policy. We derive a closed-form expression of the entropy of such policies. Our formula is computationally efficient and only depends on first-order derivatives and vector products. Empirical results show that S$^2$AC yields more optimal solutions to the MaxEnt objective than SQL and SAC in the multi-goal environment, and outperforms SAC and SQL on the MuJoCo benchmark. Our code is available at: https://github.com/SafaMessaoud/S2AC-Energy-Based-RL-with-Stein-Soft-Actor-Critic

## 1 Introduction

MaxEnt RL (Todorov, 2006; Ziebart, 2010; Haarnoja et al., 2017; Kappen, 2005; Toussaint, 2009; Theodorou et al., 2010; Abdolmaleki et al., 2018; Haarnoja et al., 2018a; Vieillard et al., 2020) has been proposed to address challenges hampering the deployment of RL to real-world applications, including stability, sample efficiency (Gu et al., 2017), and robustness (Eysenbach & Levine, 2022). Instead of learning a deterministic policy, as in classical RL (Sutton et al., 1999; Schulman et al., 2017; Silver et al., 2014; Lillicrap et al., 2015), MaxEnt RL learns a stochastic policy that captures the intricacies of the action space. This enables better exploration during training and eventually better robustness to environmental perturbations at

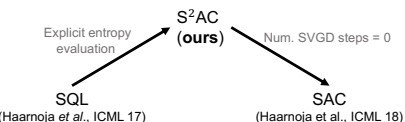

Figure 1: Comparing S$^2$AC to SQL and SAC. S$^2$AC with a parameterized policy is reduced to SAC if the number of SVGD steps is 0. SQL becomes equivalent to S$^2$AC if the entropy is evaluated explicitly with our derived formula.

test time, *i.e.*, the agent learns multimodal action space distributions which enables picking the next best action in case a perturbation prevents the execution of the optimal one. To achieve this, MaxEnt RL models the policy using the expressive family of EBMs (LeCun et al., 2006). This translates into learning policies that maximize the sum of expected future reward and expected future entropy. However, estimating the entropy of such complex distributions remains an open problem.

To address this, existing approaches either use tricks to go around the entropy computation or make limiting assumptions on the policy. This results in either poor scalability or convergence to suboptimal solutions. For example, SQL (Haarnoja et al., 2017) implicitly incorporates entropy in the Q-function computation. This requires using importance sampling, which results in high variability and hence poor training stability and limited scalability to high dimensional action spaces. SAC (Haarnoja

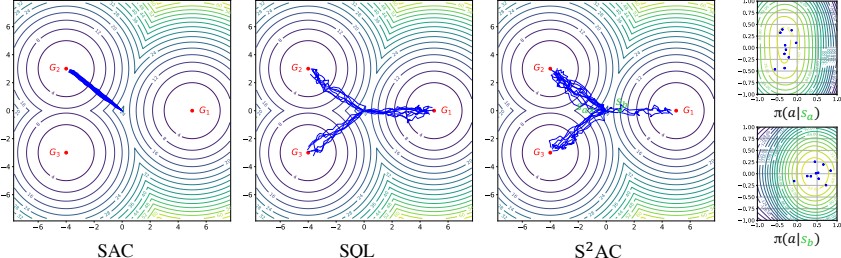

Figure 2: $S^2$AC learns a more optimal solution to the MaxEnt RL objective than SAC and SQL. We design a multigoal environment where an agent starts from the center of the 2-d map and tries to reach one of the three goals ($G_1$, $G_2$, and $G_3$). The maximum expected future reward (level curves) is the same for all the goals but the expected future entropy is different (higher on the path to $G_2/G_3$): the action distribution $\pi(a|s)$ is bi-modal on the path to the left ($G_2$ and $G_3$) and unimodal to the right ($G_1$). Hence, we expect the optimal policy for the MaxEnt RL objective to assign more weights to $G_2$ and $G_3$. We visualize trajectories (in blue) sampled from the policies learned using SAC, SQL, and $S^2$AC. SAC quickly commits to a single mode due to its actor being tied to a Gaussian policy. Though SQL also recovers the three modes, the trajectories are evenly distributed. $S^2$AC recovers all the modes and approaches the left two goals more frequently. This indicates that it successfully maximizes not only the expected future reward but also the expected future entropy.

et al., 2018a), on the other hand, follows a variational inference procedure by fitting a Gaussian distribution to the EBM policy. This enables a closed-form evaluation of the entropy but results in a suboptimal solution. For instance, SAC fails in environments characterized by multimodal action distributions. Similar to SAC, IAPO (Marino et al., 2021) models the policy as a uni-modal Gaussian. Instead of optimizing a MaxEnt objective, it achieves multimodal policies by learning a collection of parameter estimates (mean, variance) through different initializations for different policies. To improve the expressiveness of SAC, SSPG (Cetin & Celiktutan, 2022) and SAC-NF (Mazoure et al., 2020) model the policy as a Markov chain with Gaussian transition probabilities and as a normalizing flow (Rezende & Mohamed, 2015), respectively. However, due to training stability issues, the reported results in Cetin & Celiktutan (2022) show that though both models learn multi-modal policies, they fail to maximize the expected future entropy in positive rewards setups.

We propose a new algorithm, $S^2$AC, that yields a more optimal solution to the MaxEnt RL objective. To achieve *expressivity*, $S^2$AC models the policy as a Stein Variational Gradient Descent (SVGD) (Liu, 2017) sampler from an EBM over Q-values (target distribution). SVGD proceeds by first sampling a set of particles from an initial distribution, and then iteratively transforming these particles via a sequence of updates to fit the target distribution. To compute a *closed-form estimate of the entropy* of such policies, we use the change-of-variable formula for pdfs (Devore et al., 2012). We prove that this is only possible due to the invertibility of the SVGD update rule, which does not necessarily hold for other popular samplers (*e.g.*, Langevin Dynamics (Welling & Teh, 2011)). While normalizing flow models (Rezende & Mohamed, 2015) are also invertible, SVGD-based policy is more expressive as it encodes the inductive bias about the unnormalized density and incorporates a dispersion term to encourage multi-modality, whereas normalizing flows encode a restrictive class of invertible transformations (with easy-to-estimate Jacobian determinants). Moreover, our formula is computationally efficient and only requires evaluating first-order derivatives and vector products. To improve *scalability*, we model the initial distribution of the SVGD sampler as an isotropic Gaussian and learn its parameters, *i.e.*, mean and standard deviation, end-to-end. We show that this results in faster convergence to the target distribution, *i.e.*, fewer SVGD steps. Intuitively, the initial distribution learns to contour the high-density region of the target distribution while the SVGD updates result in better and faster convergence to the modes within that region. Hence, our approach is as parameter efficient as SAC, since the SVGD updates do not introduce additional trainable parameters.

Note that $S^2$AC can be reduced to SAC when the number of SVGD steps is zero. Also, SQL becomes equivalent to $S^2$AC if the entropy is computed explicitly using our formula (the policy in SQL is an amortized SVGD sampler). Beyond RL, the backbone of $S^2$AC is a new variational inference algorithm with a more expressive and scalable distribution characterized by a closed-form entropy estimate. We believe that this variational distribution can have a wider range of exciting applications.

We conduct extensive empirical evaluations of $S^2$AC from three aspects. We start with a sanity check on the merit of our derived SVGD-based entropy estimate on target distributions with known entropy values (*e.g.*, Gaussian) or log-likelihoods (*e.g.*, Gaussian Mixture Models) and assess its

sensitivity to different SVGD parameters (kernel, initial distribution, number of steps and number of particles). We observe that its performance depends on the choice of the kernel and is robust to variations of the remaining parameters. In particular, we find out that the kernel should be chosen to guarantee inter-dependencies between the particles, which turns out to be essential for invertibility. Next, we assess the performance of S$^2$AC on a multi-goal environment (Haarnoja et al., 2017) where different goals are associated with the same positive (maximum) expected future reward but different (maximum) expected future entropy. We show that S$^2$AC learns multimodal policies and effectively maximizes the entropy, leading to better robustness to obstacles placed at test time. Finally, we test S$^2$AC on the MuJoCo benchmark (Duan et al., 2016). S$^2$AC yields better performances than the baselines on four out of the five environments. Moreover, S$^2$AC shows higher sample efficiency as it tends to converge with fewer training steps. These results were obtained from running SVGD for only three steps, which results in a small overhead compared to SAC during training. Furthermore, to maximize the run-time efficiency during testing, we train an amortized SVGD version of the policy to mimic the SVGD-based policy. Hence, this reduces inference to a forward pass through the policy network without compromising the performance.

## 2 PRELIMINARIES

### 2.1 SAMPLERS FOR ENERGY-BASED MODELS

In this work, we study three representative methods for sampling from EBMs: (1) Stochastic Gradient Langevin Dynamics (SGLD) & Deterministic Langevin Dynamics (DLD) (Welling & Teh, 2011), (2) Hamiltonian Monte Carlo (HMC) (Neal et al., 2011), and (3) Stein Variational Gradient Descent (SVGD) (Liu & Wang, 2016). We review SVGD here since it is the sampler we eventually use in S$^2$AC, and leave the rest to Appendix C.1. SVGD is a particle-based Bayesian inference algorithm. Compared to SGLD and HMC which have a single particle in their dynamics, SVGD operates on a set of particles. Specifically, SVGD samples a set of $m$ particles $\{a_j\}_{j=1}^m$ from an initial distribution $q^0$ which it then transforms through a sequence of updates to fit the target distribution. Formally, at every iteration $l$, SVGD applies a form of functional gradient descent $\Delta f$ that minimizes the KL-divergence between the target distribution $p$ and the proposal distribution $q^l$ induced by the particles, *i.e.*, the update rule for the $i^{\text{th}}$ particles is: $a_i^{l+1} = a_i^l + \epsilon \Delta f(a_i^l)$ with

$$\Delta f(a_i^l) = \mathbb{E}_{a_j^l \sim q^l} \left[ k(a_i^l, a_j^l) \nabla_{a_j^l} \log p(a_j^l) + \nabla_{a_j^l} k(a_i^l, a_j^l) \right]. \tag{1}$$

Here, $\epsilon$ is the step size and $k(\cdot, \cdot)$ is the kernel function, *e.g.*, the RBF kernel: $k(a_i, a_j) = \exp(||a_i - a_j||^2/2\sigma^2)$. The first term within the gradient drives the particles toward the high probability regions of $p$, while the second term serves as a repulsive force to encourage dispersion.

### 2.2 MAXIMUM-ENTROPY RL

We consider an infinite horizon Markov Decision Process (MDP) defined by a tuple $(\mathcal{S}, \mathcal{A}, p, r)$, where $\mathcal{S}$ is the state space, $\mathcal{A}$ is the action space and $p : \mathcal{S} \times \mathcal{A} \times \mathcal{S} \to [0, \infty]$ is the state transition probability modeling the density of the next state $s_{t+1} \in \mathcal{S}$ given the current state $s_t \in \mathcal{S}$ and action $a_t \in \mathcal{A}$. Additionally, we assume that the environment emits a bounded reward function $r \in [r_{\min}, r_{\max}]$ at every iteration. We use $\rho_\pi(s_t)$ and $\rho_\pi(s_t, a_t)$ to denote the state and state-action marginals of the trajectory distribution induced by a policy $\pi(a_t|s_t)$. We consider the setup of continuous action spaces Lazaric et al. (2007); Lee et al. (2018); Zhou & Lu (2023). MaxEnt RL (Todorov, 2006; Ziebart, 2010; Rawlik et al., 2012) learns a policy $\pi^*(a_t|s_t)$, that instead of maximizing the expected future reward, maximizes the sum of the expected future reward and entropy:

$$\pi^* = \arg\max_\pi \sum_t \gamma^t \mathbb{E}_{(s_t, a_t) \sim \rho_\pi} \left[ r(s_t, a_t) + \alpha \mathcal{H}(\pi(\cdot|s_t)) \right], \tag{2}$$

where $\alpha$ is a temperature parameter controlling the stochasticity of the policy and $\mathcal{H}(\pi(\cdot|s_t))$ is the entropy of the policy at state $s_t$. The conventional RL objective can be recovered for $\alpha = 0$. Note that the MaxEnt RL objective above is equivalent to approximating the policy, modeled as an EBM over Q-values, by a variational distribution $\pi(a_t|s_t)$ (see proof of equivalence in Appendix D), *i.e.*,

$$\pi^* = \arg\min_\pi \sum_t \mathbb{E}_{s_t \sim \rho_\pi} \left[ D_{KL}\left( \pi(\cdot|s_t) \| \exp(Q(s_t, \cdot)/\alpha)/Z \right) \right], \tag{3}$$

where $D_{KL}$ is the KL-divergence and $Z$ is the normalizing constant. We now review two landmark MaxEnt RL algorithms: SAC (Haarnoja et al., 2018a) and SQL (Haarnoja et al., 2017).

**SAC** is an actor-critic algorithm that alternates between policy evaluation, *i.e.*, evaluating the Q-values for a policy $\pi_\theta(a_t|s_t)$:

$$Q_\phi(s_t, a_t) \leftarrow r(s_t, a_t) + \gamma \mathbb{E}_{s_{t+1}, a_{t+1} \sim \rho_{\pi_\theta}} \left[ Q_\phi(s_{t+1}, a_{t+1}) + \alpha \mathcal{H}(\pi_\theta(\cdot|s_{t+1})) \right] \tag{4}$$

and policy improvement, *i.e.*, using the updated Q-values to compute a better policy:

$$\pi_\theta = \arg\max_\theta \sum_t \mathbb{E}_{s_t, a_t \sim \rho_{\pi_\theta}} \left[ Q_\phi(a_t, s_t) + \alpha \mathcal{H}(\pi_\theta(\cdot|s_t)) \right]. \tag{5}$$

SAC models $\pi_\theta$ as an isotropic Gaussian, *i.e.*, $\pi_\theta(\cdot|s) = \mathcal{N}(\mu_\theta, \sigma_\theta I)$. While this enables computing a closed-form expression of the entropy, it incurs an over-simplification of the true action distribution, and thus cannot represent complex distributions, *e.g.*, multimodal distributions.

**SQL** goes around the entropy computation, by defining a soft version of the value function $V_\phi = \alpha \log \left( \int_\mathcal{A} \exp \left( \frac{1}{\alpha} Q_\phi(s_t, a') \right) da' \right)$. This enables expressing the Q-value (Eq (4)) independently from the entropy, *i.e.*, $Q_\phi(s_t, a_t) = r(s_t, a_t) + \gamma \mathbb{E}_{s_{t+1} \sim p}[V_\phi(s_{t+1})]$. Hence, SQL follows a soft value iteration which alternates between the updates of the "soft" versions of $Q$ and value functions:

$$Q_\phi(s_t, a_t) \leftarrow r(s_t, a_t) + \gamma \mathbb{E}_{s_{t+1} \sim p}[V_\phi(s_{t+1})], \; \forall(s_t, a_t) \tag{6}$$

$$V_\phi(s_t) \leftarrow \alpha \log \left( \int_\mathcal{A} \exp \left( \frac{1}{\alpha} Q_\phi(s_t, a') \right) da' \right), \; \forall s_t. \tag{7}$$

Once the $Q_\phi$ and $V_\phi$ functions converge, SQL uses amortized SVGD Wang & Liu (2016) to learn a stochastic sampling network $f_\theta(\xi, s_t)$ that maps noise samples $\xi$ into the action samples from the EBM policy distribution $\pi^*(a_t|s_t) = \exp \left( \frac{1}{\alpha}(Q^*(s_t, a_t) - V^*(s_t)) \right)$. The parameters $\theta$ are obtained by minimizing the loss $J_\theta(s_t) = D_{KL}\left( \pi_\theta(\cdot|s_t) \| \exp\left( \frac{1}{\alpha}(Q_\phi^*(s_t, \cdot) - V_\phi^*(s_t)) \right) \right)$ with respect to $\theta$. Here, $\pi_\theta$ denotes the policy induced by $f_\theta$. SVGD is designed to minimize such KL-divergence without explicitly computing $\pi_\theta$. In particular, SVGD provides the most greedy direction as a functional $\Delta f_\theta(\cdot, s_t)$ (Eq (1)) which can be used to approximate the gradient $\partial J_\theta / \partial a_t$. Hence, the gradient of the loss $J_\theta$ with respect to $\theta$ is: $\partial J_\theta(s_t)/\partial \theta \propto \mathbb{E}_\xi \left[ \Delta f_\theta(\xi, s_t) \partial f_\theta(\xi, s_t)/\partial \theta \right]$. Note that the integral in Eq (7) is approximated via importance sampling, which is known to result in high variance estimates and hence poor scalability to high dimensional action spaces. Moreover, amortized generation is usually unstable and prone to mode collapse, an issue similar to GANs. Therefore, SQL is outperformed by SAC Haarnoja et al. (2018a) on benchmark tasks like MuJoCo.

## 3 APPROACH

We introduce $S^2$AC, a new actor-critic MaxEnt RL algorithm that uses SVGD as the underlying actor to generate action samples from policies represented using EBMs. This choice is motivated by the expressivity of distributions that can be fitted via SVGD. Additionally, we show that we can derive a closed-form entropy estimate of the SVGD-induced distribution, thanks to the invertibility of the update rule, which does not necessarily hold for other EBM samplers. Besides, we propose a parameterized version of SVGD to enable scalability to high-dimensional action spaces and non-smooth Q-function landscapes. $S^2$AC is hence capable of learning a more optimal solution to the MaxEnt RL objective (Eq (2)) as illustrated in Figure 2.

### 3.1 STEIN SOFT ACTOR CRITIC

Like SAC, $S^2$AC performs soft policy iteration which alternates between policy evaluation and policy improvement. The difference is that we model the actor as a *parameterized sampler from an EBM*. Hence, the policy distribution corresponds to an expressive EBM as opposed to a Gaussian.

**Critic.** The critic's parameters $\phi$ are obtained by minimizing the Bellman loss as traditionally:

$$\phi^* = \arg\min_\phi \mathbb{E}_{(s_t, a_t) \sim \rho_{\pi_\theta}} \left[ (Q_\phi(s_t, a_t) - \hat{y})^2 \right], \tag{8}$$

with the target $\hat{y} = r_t(s_t, a_t) + \gamma \mathbb{E}_{(s_{t+1}, a_{t+1}) \sim \rho_\pi} \left[ Q_{\bar\phi}(s_{t+1}, a_{t+1}) + \alpha \mathcal{H}(\pi(\cdot|s_{t+1})) \right]$. Here $\bar\phi$ is an exponentially moving average of the value network weights (Mnih et al., 2015).

**Actor as an EBM sampler.** The actor is modeled as a sampler from an EBM over the Q-values. To generate a set of valid actions, the actor first samples a set of particles $\{a^0\}$ from an initial distribution $q^0$ (*e.g.*, Gaussian). These particles are then updated over several iterations $l \in [1, L]$, *i.e.*, $\{a^{l+1}\} \leftarrow \{a^l\} + \epsilon h(\{a^l\}, s)$ following the sampler dynamics characterized by a transformation $h$ (*e.g.*, for SVGD, $h = \Delta f$ in Eq (1)). If $q^0$ is tractable and $h$ is invertible, it's possible to compute a closed-form expression of the distribution of the particles at the $l^{\text{th}}$ iteration via the change of variable formula Devore et al. (2012): $q^l(a^l|s) = q^{l-1}(a^{l-1}|s) \left| \det(I + \epsilon \nabla_{a^l} h(a^l, s)) \right|^{-1}, \forall l \in [1, L]$. In this case, the policy is represented using the particle distribution at the final step $L$ of the sampler dynamics, *i.e.*, $\pi(a|s) = q^L(a^L|s)$ and the entropy can be estimated by averaging $\log q^L(a^L|s)$ over a set of particles (Section 3.2). We study the invertibility of popular EBM samplers in Section 3.3.

**Parameterized initialization.** To reduce the number of steps required to converge to the target distribution (hence reducing computation cost), we further propose modeling the initial distribution as a parameterized isotropic Gaussian, *i.e.*, $a^0 \sim \mathcal{N}(\mu_\theta(s), \sigma_\theta(s))$. The parameterization trick is then used to express $a^0$ as a function of $\theta$. Intuitively, the actor would learn $\theta$ such that the initial distribution is close to the target distribution. Hence, fewer steps are required to converge, as illustrated in Figure 3. Note that if the number of steps $L = 0$, S²AC is

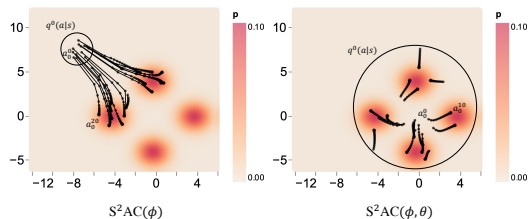

Figure 3: S²AC($\phi, \theta$) achieves faster convergence to the target distribution (in orange) than S²AC($\phi$) by parameterizing the initial distribution $\mathcal{N}(\mu_\theta, \sigma_\theta)$ of the SVGD sampler.

reduced to SAC. Besides, to deal with the non-smooth nature of deep Q-function landscapes which might lead to particle divergence in the sampling process, we bound the particle updates to be within a few standard deviations ($t$) from the mean of the learned initial distribution, *i.e.*, $-t\sigma_\theta \leq a_\theta^l \leq t\sigma_\theta$, $\forall l \in [1, L]$. Eventually, the initial distribution $q_\theta^0$ learns to contour the high-density region of the target distribution and the following updates refine it by converging to the spanned modes. Formally, the parameters $\theta$ are computed by minimizing the expected KL-divergence between the policy $q_\theta^L$ induced by the particles from the sampler and the EBM of the Q-values:

$$\theta^* = \arg\max_\theta \mathbb{E}_{s_t \sim \mathcal{D}, a_\theta^L \sim \pi_\theta} \left[ Q_\phi(s_t, a_\theta^L) \right] + \alpha \mathbb{E}_{s_t \sim \mathcal{D}} \left[ \mathcal{H}(\pi_\theta(\cdot|s_t)) \right]$$

$$\text{s.t.} \quad -t\sigma_\theta \leq a_\theta^l \leq t\sigma_\theta, \quad \forall l \in [1, L]. \tag{9}$$

Here, $\mathcal{D}$ is the replay buffer. The derivation is in Appendix E. Note that the constraint does not truncate the particles as it is not an invertible transformation which then violates the assumptions of the change of variable formula. Instead, we sample more particles than we need and select the ones that stay within the range. We call S²AC($\phi, \theta$) and S²AC($\phi$) as two versions of S²AC with/without the parameterized initial distribution. The complete S²AC algorithm is in Algorithm 1 of Appendix A.

### 3.2 A CLOSED-FORM EXPRESSION OF THE POLICY'S ENTROPY

A critical challenge in MaxEnt RL is how to efficiently compute the entropy term $\mathcal{H}(\pi(\cdot|s_{t+1}))$ in Eq (2). We show that, if we model the policy as an iterative sampler from the EBM, under certain conditions, we can derive a closed-form estimate of the entropy at convergence.

**Theorem 3.1.** *Let $F : \mathbb{R}^n \to \mathbb{R}^n$ be an invertible transformation of the form $F(a) = a + \epsilon h(a)$. We denote by $q^L(a^L)$ the distribution obtained from repeatedly applying $F$ to a set of samples $\{a^0\}$ from an initial distribution $q^0(a^0)$ over $L$ steps, i.e., $a^L = F \circ F \circ \cdots \circ F(a^0)$. Under the condition $\epsilon ||\nabla_{a_i^l} h(a_i)||_\infty \ll 1, \forall l \in [1, L]$, the distribution of the particles at the $L^{th}$ step is:*

$$\log q^L(a^L) \approx \log q^0(a^0) - \epsilon \sum_{l=0}^{L-1} \text{Tr}(\nabla_{a^l} h(a^l)) + \mathcal{O}(\epsilon^2 dL). \tag{10}$$

*Here, $d$ is the dimensionality of $a$, i.e., $a \in \mathbb{R}^d$ and $\mathcal{O}(\epsilon^2 dL)$ is the order of approximation error.*
*Proof Sketch:* As $F$ is invertible, we apply the change of variable formula (Appendix C.2) on the transformation $F \circ F \circ \cdots F$ and obtain: $\log q^L(a^L) = \log q^0(a^0) - \sum_{l=0}^{L-1} \log |\det(I + \epsilon \nabla_{a^l} h(a^l))|$. Under the assumption $\epsilon ||\nabla_{a_i} h(a_i)||_\infty \ll 1$, we apply the corollary of Jacobi's formula (Appendix C.3) and get Eq. (10). The detailed proof is in Appendix F. Note that the condition $\epsilon ||\nabla_{a_i} h(a_i)||_\infty \ll 1$ can always be satisfied when we choose a sufficiently small step size $\epsilon$, *or* the gradient of $h(a)$ is small, *i.e.*, $h(a)$ is Lipschitz continuous with a sufficiently small constant.
It follows from the theorem above, that the entropy of a policy modeled as an EBM sampler (Eq (9)) can be expressed analytically as:

$$\mathcal{H}(\pi_\theta(\cdot|s)) = -\mathbb{E}_{a_\theta^0 \sim q_\theta^0} \left[ \log q_\theta^L(a_\theta^L|s) \right] \approx -\mathbb{E}_{a_\theta^0 \sim q_\theta^0} \left[ \log q_\theta^0(a^0|s) - \epsilon \sum_{l=0}^{L-1} \text{Tr}\left( \nabla_{a_\theta^l} h(a_\theta^l, s) \right) \right]. \tag{11}$$

In the following, we drop the dependency of the action on $\theta$ for simplicity of the notation.

### 3.3 INVERTIBLE POLICIES

Next, we study the invertibility of three popular EBM samplers: SVGD, SGLD, and HMC as well as the efficiency of computing the trace, *i.e.*, $\text{Tr}(\nabla_{a^l} h(a^l, s))$ in Eq (10) for the ones that are invertible.
**Proposition 3.2** (SVGD invertibility). *Given the SVGD learning rate $\epsilon$ and RBF kernel $k(\cdot, \cdot)$ with variance $\sigma$, if $\epsilon \ll \sigma$, the update rule of SVGD dynamics defined in Eq (1) is invertible.*

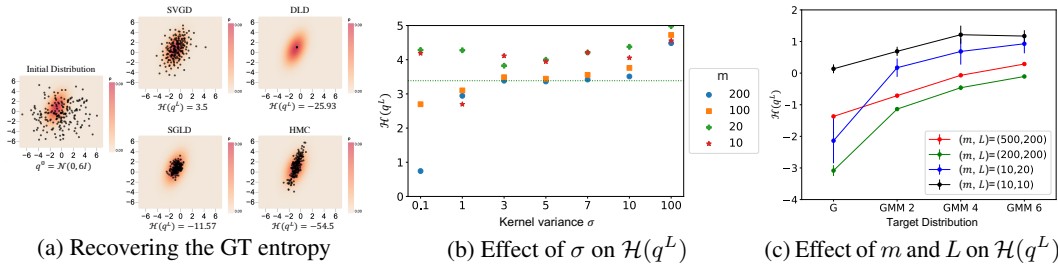

(a) Recovering the GT entropy     (b) Effect of $\sigma$ on $\mathcal{H}(q^L)$     (c) Effect of $m$ and $L$ on $\mathcal{H}(q^L)$

Figure 4: Entropy evaluation results.

*Proof Sketch:* We use the explicit function theorem to show that the Jacobian $\nabla_a F(a, s)$ of the update rule $F(a, s)$ is diagonally dominated and hence invertible. This yields invertibility of $F(a, s)$. See detailed proof in Appendix G.3.

**Theorem 3.3.** *The closed-form estimate of $\log q^L(a^L|s)$ for the SVGD based sampler with an RBF kernel $k(\cdot, \cdot)$ is*

$$\log q^L(a^L|s) \approx \log q^0(a^0|s) + \frac{\epsilon}{m\sigma^2} \sum_{l=0}^{L-1} \sum_{j=1, a^l \neq a_j^l}^{m} k(a_j^l, a^l)\Big((a^l - a_j^l)^\top \nabla_{a_j^l} Q(s, a_j^l) + \frac{\alpha}{\sigma^2}\|a^l - a_j^l\|^2 - d\alpha\Big).$$

Here, $(\cdot)^\top$ denotes the transpose of a matrix/vector. Note that the entropy does not depend on any matrix computation, but only on vector dot products and first-order vector derivatives. The proof is in Appendix H.1. Intuitively, the derived likelihood is proportional to (1) the concavity of the curvature of the Q-landscape, captured by a weighted average of the neighboring particles' Q-value gradients and (2) pairwise-distances between the neighboring particles ($\sim \|a_i^l - a_j^l\|^2 \cdot \exp(\|a_i^l - a_j^l\|^2)$), *i.e.*, the larger the distance the higher is the entropy. We elaborate on the connection between this formula and non-parametric entropy estimators in Appendix B.

**Proposition 3.4** (SGLD, HMC). *The SGLD and HMC updates are not invertible w.r.t. $a$.*

*Proof Sketch:* SGLD is stochastic (noise term) and thus not injective. HMC is only invertible if conditioned on the velocity $v$. Detailed proofs are in Appendices G.1-G.2.

From the above theoretic analysis, we can see that SGLD update is not invertible and hence is not suitable as a sampler for $S^2AC$. While the HMC update is invertible, its derived closed-form entropy involves calculating Hessian and hence computationally more expensive. Due to these considerations, we choose to use SVGD with an RBF kernel as the underlying sampler of $S^2AC$.

## 4 RESULTS

We first evaluate the correctness of our proposed closed-form entropy formula. Then we present the results of different RL algorithms on multigoal and MuJoCo environments.

### 4.1 ENTROPY EVALUATION

This experiment tests the correctness of our entropy formula. We compare the estimated entropy for distributions (with known ground truth entropy or log-likelihoods) using different samplers and study the sensitivity of the formula to different samplers' parameters. **(1) Recovering the ground truth entropy.** In Figure 4a, we plot samples (black dots) obtained by SVGD, SGLD, DLD and HMC at convergence to a Gaussian with ground truth entropy $\mathcal{H}(p) = 3.41$, starting from the same initial distribution (leftmost sub-figure). We also report the entropy values computed via Eq.(11). Unlike SGLD, DLD, and HMC, SVGD recovers the ground truth entropy. This empirically supports Proposition 3.4 that SGLD, DLD, and HMC are not invertible. **(2) Effect of the kernel variance.** Figure 4b shows the effect of different SVGD kernel variances $\sigma$, where we use the same initial Gaussian from Figure 4a. We also visualize the particle distributions after $L$ SVGD steps for the different configurations in Figure 9 of Appendix I. We can see that when the kernel variance is too small (*e.g.*, $\sigma = 0.1$), the invertibility is violated, and thus the estimated entropy is wrong even at convergence. On the other extreme when the kernel variance is too large (*e.g.*, $\sigma = 100$), *i.e.*, when the particles are too scattered initially, the particles do not converge to the target Gaussian due to noisy gradients in the first term of Eq.(1). The best configurations hence lie somewhere in between (*e.g.*, $\sigma \in \{3, 5, 7\}$). **(3) Effect of SVGD steps and particles.** Figure 4c and Figure 10b (Appendix. I) show the behavior of our entropy formula under different configurations of the number of SVGD steps and particles, on two settings: (i) GMM $M$ with an increasing number of components $M$, and (ii) distributions with increasing ground truth entropy values, *i.e.*, Gaussians with increasing variances $\sigma$. Results show that our entropy consistently grows with an increasing $M$ (Figure 4c) and increasing $\sigma$ (Figure 10b), even when a small number of SVGD steps and particles is used (*e.g.*, $L = 10, m = 10$).

## 4.2 MULTI-GOAL EXPERIMENTS

To check if $S^2AC$ learns a better solution to the max-entropy objective (Eq (2)), we design a new multi-goal environment as shown in Figure 5. The agent is a 2D point mass at the origin trying to reach one of the goals (in red). Q-landscapes are depicted by level curves. Actions are bounded in $[-1, 1]$ along both axes. Critical states for the analysis are marked with blue crosses. It is built on the multi-goal environment in Haarnoja et al. (2017) with modifications such that all the goals have (i) the same maximum expected future reward (positive) but (ii) different maximum expected future entropy. This is achieved by asymmetrically placing the goals (two goals on the left side and one on the right, leading to a higher

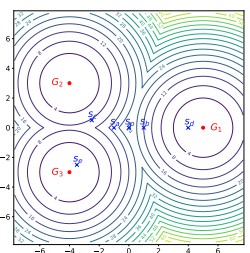

Figure 5: Multigoal env.

expected future entropy on the left side) while assigning the same final rewards to all the goals. The problem setup and hyperparameters are detailed in Appendix J. **(1) Multi-modality.** Figure 6 visualizes trajectories (blue lines) collected from 20 episodes of $S^2AC(\phi, \theta)$, $S^2AC(\phi)$, SAC, SQL and SAC-NF (SAC with a normalizing flow policy, Mazoure et al. (2020)) agents (rows) at test time for increasing entropy weights $\alpha$ (columns). $S^2AC$ and SQL consistently cover all the modes for all $\alpha$ values, while this is only achieved by SAC and SAC-NF for large $\alpha$ values. Note that, in the case of SAC, this comes at the expense of accuracy. Although normalizing flows are expressive enough in theory, they are known to quickly collapse to local optima in practice Kobyzev et al. (2020). The dispersion term in $S^2AC$ encodes an inductive bias to mitigate this issue. **(2) Maximizing the expected future entropy.** We also see that with increasing $\alpha$, more $S^2AC$ and SAC-NF trajectories converge to the left goals ($G_2/G_3$). This shows both models learn to maximize the expected future entropy. This is not the case for SQL whose trajectory distribution remains uniform across the goals. SAC results do not show a consistent trend. This validates the hypothesis that the entropy term in SAC only helps exploration but does not lead to maximizing future entropy. The quantified distribution over reached goals is in Figure 12 of Appendix J. **(3) Robustness/adaptability.** To assess the robustness of the learned policies, we place an obstacle (red bar in Figure 7) on the path to $G_2$. We show the test time trajectories of 20 episodes using $S^2AC$, SAC, SQL and SAC-NF agents trained with different $\alpha$'s. We observe that, for $S^2AC$ and SAC-NF, with increasing $\alpha$, more trajectories reach the goal after hitting the obstacles. This is not the case for SAC, where many trajectories hit the obstacle without reaching the goal. SQL does not manage to escape the barrier even with higher $\alpha$. Additional results on the **(4) effect of parameterization of** $q^0$, and the **(5) entropy's effect on the learned Q-landscapes** are respectively reported in Figure 11 and Figure 14 of Appendix J.

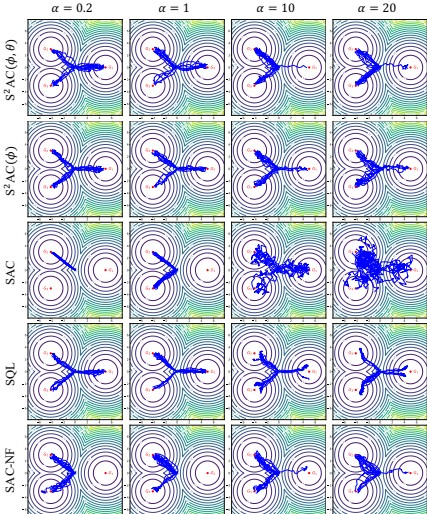

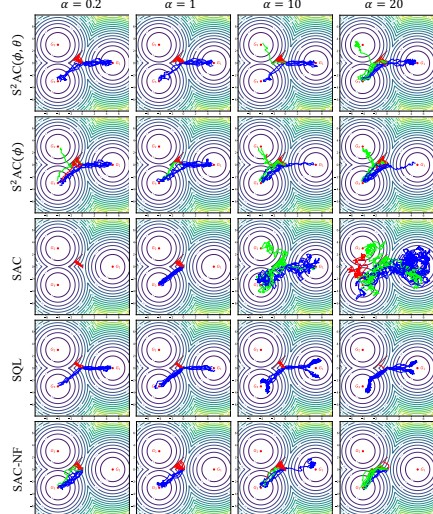

Figure 6: $S^2AC$ and SAC-NF learn to maximize the expected future entropy (biased towards $G_2/G_3$) while SAC and SQL do not. $S^2AC$ consistently recovers all modes, while SAC-NF with smaller $\alpha$'s does not, indicating its instability.

Figure 7: $S^2AC$ and SAC-NF are more robust to perturbations. Obstacle $O$ is placed diagonally at $[-1, 1]$. Trajectories that did and did not reach the goal after hitting $O$ are in green and red, respectively.

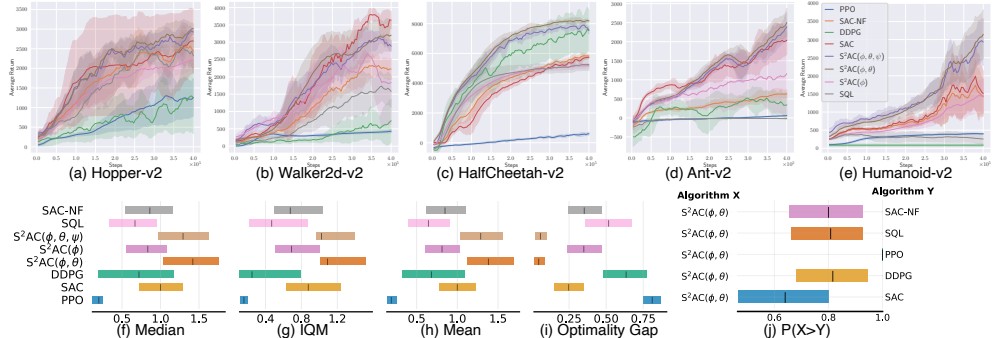

Figure 8: **(a)-(e)**: Performance curves on the MuJoCo benchmark (training). $S^2AC$ outperforms SQL and SAC-NF on all environments and SAC on 4 out of 5 environments. **(f)-(i)**: Comparison of Median, IQM, Mean, and Optimality Gap between $S^2AC$ and baseline algorithms. **(j)**: The probabilities of $S^2AC$ outperforming baseline algorithms.

## 4.3 MUJOCO EXPERIMENTS

We evaluate $S^2AC$ on five environments from MuJoCo (Brockman et al., 2016): Hopper-v2, Walker2d-v2, HalfCheetah-v2, Ant-v2, and Humanoid-v2. As baselines, we use (1) DDPG (Gu et al., 2017), (2) PPO (Schulman et al., 2015), (3) SQL (Haarnoja et al., 2017), (4) SAC-NF (Mazoure et al., 2020), and (5) SAC (Haarnoja et al., 2018a). Hyperparameters are in Appendix K.

**(1) Performance and sample efficiency.** We train five different instances of each algorithm with different random seeds, with each performing 100 evaluation rollouts every 1000 environment steps. Performance results are in Figure 8(a)-(e). The solid curves correspond to the mean returns over the five trials and the shaded region represents the minimum and maximum. $S^2AC(\phi, \theta)$ is consistently better than SQL and SAC-NF across all the environments and has superior performance than SAC in four out of five environments. Results also show that the initial parameterization was key to ensuring the scalability ($S^2AC(\phi)$ has poor performance compared to $S^2AC(\phi, \theta)$). Figure 8(f)-(j) demonstrate the statistical significance of these gains by leveraging statistics from the rliable library (Agarwal et al., 2021) which we detail in Appendix K.

**(2) Run-time.** We report the run-time of action selection of SAC, SQL, and $S^2AC$ algorithms in Table 1. $S^2AC(\phi, \theta)$ run-time increases linearly with the action space. To improve the scalability, we train an amortized version that we deploy at test-time, following (Haarnoja et al., 2017). Specifically, we train a feed-forward deepnet $f_\psi(s, z)$ to mimic the SVGD dynamics during testing, where $z$ is a random vector that allows mapping the same state to different particles. Note that we cannot use $f_\psi(s, z)$ during training

|  | Hopper | Walker2d | HalfCheetah | Ant |
|---|---|---|---|---|
| Action dim | 3 | 6 | 6 | 8 |
| State dim | 11 | 17 | 17 | 111 |
| SAC | 0.723 | 0.714 | 0.731 | 0.708 |
| SQL | 0.839 | 0.828 | 0.815 | 0.836 |
| $S^2AC(\phi, \theta)$ | 3.267 | 4.622 | 4.583 | 5.917 |
| $S^2AC(\phi, \theta, \psi)$ | 0.850 | 0.817 | 0.830 | 0.837 |

Table 1: Action selection run-time on MuJoCo.

as we need to estimate the entropy in Eq (11), which depends on the unrolled SVGD dynamics (details in Appendix K). The amortized version $S^2AC(\phi, \theta, \psi)$ has a similar run-time to SAC and SQL with a slight tradeoff in performance (Figure 8).

## 5 RELATED WORK

**MaxEnt RL** (Todorov, 2006; Ziebart, 2010; Rawlik et al., 2012) aims to learn a policy that gets high rewards while acting as randomly as possible. To achieve this, it maximizes the sum of expected future reward and expected future entropy. It is different from entropy regularization (Schulman et al., 2015; O'Donoghue et al., 2016; Schulman et al., 2017) which maximizes entropy at the current time step. It is also different from multi-modal RL approaches (Tang & Agrawal, 2018) which recover different modes with equal frequencies without considering their future entropy. MaxEnt RL has been broadly incorporated in various RL domains, including inverse RL (Ziebart et al., 2008; Finn et al., 2016), stochastic control (Rawlik et al., 2012; Toussaint, 2009), guided policy search (Levine & Koltun, 2013), and off-policy learning (Haarnoja et al., 2018a;b). MaxEnt RL is shown to maximize a lower bound of the robust RL objective (Eysenbach & Levine, 2022) and is hence less sensitive

to perturbations in state and reward functions. From the variational inference lens, MaxEnt RL aims to find the policy distribution that minimizes the *KL*-divergence to an EBM over Q-function. The desired family of variational distributions is (1) expressive enough to capture the intricacies of the Q-value landscape (*e.g.*, multimodality) and (2) has a tractable entropy estimate. These two requirements are hard to satisfy. SAC (Haarnoja et al., 2018a) uses a Gaussian policy. Despite having a tractable entropy, it fails to capture arbitrary Q-value landscapes. SAC-GMM (Haarnoja, 2018) extends SAC by modeling the policy as a Gaussian Mixture Model, but it requires an impractical grid search over the number of components. Other extensions include IAPO (Marino et al., 2021) which also models the policy as a uni-modal Gaussian but learns a collection of parameter estimates (mean, variance) through different initializations. While this yields multi-modality, it does not optimize a MaxEnt objective. SSPG (Cetin & Celiktutan, 2022) and SAC-NF (Mazoure et al., 2020) respectively improve the policy expressivity by modeling the policy as a Markov chain with Gaussian transition probabilities and as a normalizing flow. Due to training instability, the reported multi-goal experiments in (Cetin & Celiktutan, 2022) show that, though both models capture multimodality, they fail to maximize the expected future entropy in positive reward setups. SQL (Haarnoja et al., 2017), on the other hand, bypasses the explicit entropy computation altogether via a soft version of value iteration. It then trains an amortized SVGD (Wang & Liu, 2016) sampler from the EBM over the learned Q-values. However, estimating soft value functions requires approximating integrals via importance sampling which is known to have high variance and poor scalability. We propose a new family of variational distributions induced by a parameterized SVGD sampler from the EBM over Q-values. Our policy is expressive and captures multi-modal distributions while being characterized by a tractable entropy estimate.

**EBMs** (LeCun et al., 2006; Wu et al., 2018) are represented as Gibbs densities $p(x) = \exp E(x)/Z$, where $E(x) \in \mathbb{R}$ is an energy function describing inter-variable dependencies and $Z = \int \exp E(x)$ is the partition function. Despite their expressiveness, EBMs are not tractable as the partition function requires integrating over an exponential number of configurations. Markov Chain Monte Carlo (MCMC) methods (Van Ravenzwaaij et al., 2018) (*e.g.*, HMC (Hoffman & Gelman, 2014), SGLD (Welling & Teh, 2011)) are frequently used to approximate the partition function via sampling. There have been recent efforts to parameterize these samplers via deepnets (Levy et al., 2017; Gong et al., 2018; Feng et al., 2017) to improve scalability. Similarly to these methods, we propose a parameterized variant of SVGD (Liu & Wang, 2016) as an EBM sampler to enable scalability to high-dimensional action spaces. Beyond sampling, we derive a closed-form expression of the sampling distribution as an estimate of the EBM. This yields a tractable estimate of the entropy. This is opposed to previous methods for estimating EBM entropy which mostly rely on heuristic approximation, lower bounds Dai et al. (2017; 2019a), or neural estimators of mutual information (Kumar et al., 2019). The idea of approximating the entropy of EBMs via MCMC sampling by leveraging the change of variable formula was first proposed in Dai et al. (2019b). The authors apply the formula to HMC and LD, which, as we show previously, violate the invertibility assumption. To go around this, they augment the EBM family with the noise or velocity variable for LD and HMC respectively. But the derived log-likelihood of the sampling distribution turns out to be –counter-intuitively– independent of the sampler's dynamics and equal to the initial distribution, which is then parameterized using a flow model (details in Appendix B.2). We show that SVGD is invertible, and hence we sample from the original EBM, so that our derived entropy is more intuitive as it depends on the SVGD dynamics.

**SVGD-augmented RL** (Liu & Wang, 2016) has been explored under other RL contexts. Liu et al. (2017) use SVGD to learn a distribution over policy parameters. While this leads to learning diverse policies, it is fundamentally different from our approach as we are interested in learning a single multi-modal policy with a closed-form entropy formula. Castanet et al. (2023); Chen et al. (2021) use SVGD to sample from multimodal distributions over goals/tasks. We go beyond sampling and use SVGD to derive a closed-form entropy formula of an expressive variational distribution.

## 6 CONCLUSION

We propose $S^2AC$, an actor-critic algorithm that yields a more optimal solution to the MaxEnt RL objective than previously proposed approaches. $S^2AC$ achieves this by leveraging a new family of variational distributions characterized by SVGD dynamics. The proposed distribution has high expressivity, *i.e.*, it is flexible enough to capture multimodal policies in high dimensional spaces, and a tractable entropy estimate. Empirical results show that $S^2AC$ learns expressive and robust policies while having superior performance than other MaxEnt RL algorithms. For future work, we plan to study the application of the proposed variational distribution to other domains and develop benchmarks to evaluate the robustness of RL agents.

## ACKNOWLEDGMENTS

Bo An is supported by the National Research Foundation Singapore and DSO National Laboratories under the AI Singapore Programme (AISGAward No: AISG2-GC-2023-009). Haipeng Chen is supported by William & Mary FRC Faculty Research Grants.

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
