# Supplementary Material

## A  SUMMARY

In this paper, we propose a new variational distribution that we use to model the actor in the context of actor-critic MaxEntr RL algorithms. Our distribution is induced by an SVGD sampler with a parametrized initial distribution (isotropic Gaussian). It enables fitting multi-modal distribution (*e.g.*, EBM) and is characterized by a closed-form entropy estimate. Hence, it addresses the major bottleneck in classical MaxEntr RL algorithms. Our derivation is based on the unique invertibility property of the SVGD sampler, which is not satisfied for other popular samplers (*e.g.*, SGLD, HMC). The key to achieving scalability was to learn the initial Gaussian distribution such that it contours the high-density region of the target distribution, by limiting particles' updates to be always within few standard deviations of the mean of this Gaussian. This resulted in better and faster exploration of the relevant regions of the target distribution. Our proposed approach $S^2AC$ is summarized in Algorithm 1.The rest of the supplementary is organized as follows:

- Appendix B provides additional related work on the entropy estimation.
- Appendix C introduces additional preliminaries on EBM samplers, the change of variable formula and the Jacobi formula.
- Appendix D provides the derivation of the optimal policy for the MaxEntr RL objective.
- Appendix E provides the derivation of the actor objective.
- Appendices F-H provide proofs for theorems related to (1) a generic closed-form expression of log-likelihood of inverible samplers, (2) discussion of samplers invertibility and (3) closed-form likelihood derivation for SVGD.
- Appendices I-K provide additional results for the (1) entropy evaluation, (2) multigoal environment, and (3) MuJoCo environments.

---

**Algorithm 1** Stein Soft Actor Critic ($S^2AC$)

---

1: Initialize parameters $\phi$, $\theta$, hyperparameter $\alpha$, and replay buffer $\mathcal{D} \leftarrow \emptyset$
2: **for** each iteration **do**
3:     **for** each environment step $t$ **do**
4:         Sample action particles $\{a\}$ from $\pi_\theta(\cdot|s_t)$
5:         Select $a_t \in \{a\}$ using exploration strategy
6:         Sample next state $s_{t+1} \sim p(s_{t+1}|s_t, a_t)$
7:         Update replay buffer $\mathcal{D} \leftarrow \mathcal{D} \cup (s_t, a_t, r_t, s_{t+1})$
8:     **for** each gradient step **do**
9:         **Critic update:**
10:           Sample particles $\{a\}$ from an EMB sampler $\pi_\theta(\cdot|s_{t+1})$
11:           Compute entropy $\mathcal{H}(\pi_\theta(\cdot|s_{t+1}))$ using Eq.(11)
12:           Update $\phi$ using Eq.(8)
13:         **Actor update:**
14:           Update $\theta$ using Eq.(9)

---

## B  ADDITIONAL RELATED-WORK

### B.1  ENTROPY

The differential entropy Cover (1999); Shannon (2001) of a $p$-dimensional random variable $X$ with a probability density function $p(x)$ is defined by: $H(p) = -\int p(x) \ln p(x) dx$. The differential entropy plays a central role in information and communication theory, statistics Tarasenko (1968), signal processing Vasicek (2015); Learned-Miller & III (2003), machine learning and pattern recognition Mannor et al. (2005); Rubinstein & Kroese (2004); Hino & Murata (2010); Liu et al. (2022); Wulfmeier et al. (2015). For example, Max-Entropy RL Wulfmeier et al. (2015); Haarnoja et al. (2017; 2018a) methods augment the expected reward objective with an entropy maximization term which results in learning multi-modal policies and more robustness. Recently, Liu *et. al* Liu et al. (2022) propose maximizing the entropy of the discriminator distribution to combat mode collapse. In statistical mechanics entropy appears as the negative of the rate function to quantify the fluctuations around thermodynamic equilibrium Roldán et al. (2021). Estimating the differential entropy for expressive distributions is a challenging problem as it requires computing a closed-form expression of the probability density function. Several non-parametric approaches Beirlant et al. (1997); Györfi & van der Meulen (1987); Paninski (2003); Pérez-Cruz (2008) based on approximating the entropy using samples $\mathcal{D} = \{x_i\}_{i=1}^{|\mathcal{D}|}$ from $p(x)$, have been proposed in the literature. These methods can be classified into (1) plug-in estimates Ahmad & Lin (1976); Ivanov & Rozhkova (1981); Joe (1989) which approximate $p(x)$ via a kernel density estimate, (2) samples spacing Beirlant & van Zuijlen (1985); Cressie (1978); Dudewicz & Van Der Meulen (1981); Hall (1986) and (3) nearest-neighbor distances based estimates Bernhofen et al. (1996); Bickel & Breiman (1983); Kozachenko & Leonenko (1987) which express the entropy in terms of pairwise distances between the samples (larger distances imply higher entropy). Next, we review the work on entropy estimation of Energy-Based-Models (EBMs).

### B.2  ENTROPY OF EBMS

In this work, we are interested in computing entropy estimates for the class of EBMs LeCun et al. (2006) represented as Gibbs densities $p(x) = \frac{\exp E(x)}{Z}$, where $E(x) \in \mathbb{R}$ is an energy function describing inter-variable dependencies and $Z = \int \exp E(x)$ is an intractable partition function. EBMs provide a unified framework for many probabilistic and non-probabilistic approaches, particularly for learning and inference in structured models and are widely used in computer science (*e.g.*, semantic segmentation, colorization, image generation, inverse optimal control, collaborative filtering) Salakhutdinov et al. (2007); Messaoud et al. (2018); Zhao et al. (2021); Gao et al. (2020); Xie et al. (2020); Zheng et al. (2021); Carleo & Troyer (2017); Messaoud et al. (2020); Xie et al. (2021a); Pang et al. (2021); Messaoud (2021); Xie et al. (2016); Xie et al.; 2021b; 2022) and physics Carleo & Troyer (2017); Gao & Duan (2017); Torlai et al. (2018); Melko et al. (2019) (*e.g.*, to model the wavefunctions of quantum systems). To estimate the entropy of EBMs, previous methods mostly rely on heuristic approximation, lower bounds Dai et al. (2017; 2019a), or neural estimators of mutual information to approximate the entropy Kumar et al. (2019). The idea of approximating the entropy of EBMs via the one from an MCMC sampler by leveraging the change of variable formula was first proposed by Dai et al. (2019b). Specifically, the authors apply the formula to HMC and LD which, as we show in Appendix. G, violate the invertibility assumption. To go around this, the authors propose augmenting the EBM family with the noise or velocity variable for, respectively, LD and HMC, *i.e.*, sampling from $p(x)$ is replaced with sampling from $p(x, v)$ or $p(x, \xi)$. The authors assume that the sampler update rule is invertible with respect to the augmented samples $(x, v)$ and $(x, \xi)$. However, computing the determinant of the update rule with respect to the augmented variable is always equal to 1 in this case. Hence, the resulting log-likelihood of the sampling distribution is, counter-intuitively, independent of the sampler's dynamics and equal to the initial distribution, *i.e.*, $\log q^L(a^L) = \log q^0(a^0)$, which the author model using a flow model. Differently, we show that SVGD is invertible, our entropy depends on the dynamics of SVGD, we still sample from the original EBM $p(x)$ and our initial distribution is a simple Gaussian. Similarly to the non-parametrized entropy estimates described above, our formula leverages pairwise distances between the neighboring samples. Differently, our formula is also based on the curvature of the energy function $E(x)$ (measured by a weighted average of neighboring particle gradients $\nabla_x E(x)$). Hence maximizing our derived entropy results in the intuitive effect of learning smoother energy landscapes.

## C ADDITIONAL PRELIMINARIES

In the following, we review (1) additional samplers for EBMs, (2) the change of variable formula and (3) the corollary of the Jacobi's formula.

### C.1 ADDITIONAL SAMPLERS FOR EBMS

**SGLD** (Welling & Teh, 2011) is a popular Markov chain Monte Carlo (MCMC) method for sampling from a distribution. It initializes a sample $a^0$ from a random distribution, and then in each step $l+1$ it adds the gradient of the current proposal distribution $p(a)$ to the previous sample $a^l$, together with a Brownian motion $\xi \sim N(0, I)$. We denote the step size as $\epsilon$. The iterative update for SGLD is:

$$a^{l+1} = a^l + \epsilon \nabla_{a^l} \log p(a^l) + \sqrt{2\epsilon} \xi. \tag{12}$$

**DLD** are equivalent to SGLD without the noise term, *i.e.*,

$$a^{l+1} = a^l + \epsilon \nabla_{a^l} \log p(a^l). \tag{13}$$

**HMC** is another popular variant of MCMC samplers. The most commonly used discretized Hamilton's equations are the leapfrog method (Neal et al., 2011). The three (half) steps of leapfrog updates in HMC are:

$$\begin{aligned} v^{l+1/2} &= v^l + (\epsilon/2) \nabla_a \log p(a^{l+1}) \\ a^{l+1} &= a^l + \epsilon v^{l+1/2} \\ v^{l+1} &= a^{l+1} + (\epsilon/2) \nabla_a \log p(a^{l+1}) \end{aligned} \tag{14}$$

Here $v^l$ is interpreted the velocity at iteration $l$ (assuming unit mass) and $a^l$ is the "location" of a sample in a distribution.

### C.2 CHANGE OF VARIABLE FORMULA

We first introduce the concept of an invertibile function.

**Definition C.1** (Invertibile transformation). Transformation $F : Z \to X$ is invertible iff $F(\cdot)$ is bijective, *i.e.*, simultaneously injective and surjective: (i) $F(\cdot)$ is injective iff for any $z, z' \in Z$, $F(z) = F(z') \Rightarrow z = z'$; (ii) $F(\cdot)$ is surjective iff for every $x \in X$, there exists some $z \in Z$ such that $F(z) = x$.

According to change of variable formula, the following holds when $F : Z \to X$ is an invertible function:

$$p_X(x) = p_Z(z) \Big| \det \frac{\partial F^{-1}(x)}{\partial x} \Big| = p_Z(z) \Big| \det \frac{\partial F(z)}{\partial z} \Big|^{-1}$$

### C.3 A COROLLARY OF JACOBI'S FORMULA

An important corollary of Jacobi's Formula (Magnus & Neudecker, 2019) states that, given an invertible matrix $A$, the following equality holds:

$$\log(\det A) = \text{Tr}(\log A) = \text{Tr}\left(\sum_{k=1}^{\infty} (-1)^{k+1} \frac{(A-I)^k}{k}\right).$$

The second equation is obtained by taking the power series of $\log A$. Hence, under the assumption $\|A - I\|_\infty \ll 1$, we obtain:

$$\log(\det A) \approx \text{tr}(A - I).$$

## D  DERIVATION OF THE MAXENT RL OPTIMAL POLICY

In this section, we prove that the solution $\pi^*$ of the MaxEnt RL objective

$$\max_\pi J(\pi) \equiv \sum_t \mathbb{E}_{(s_t,a_t)\sim\rho_\pi}\Big[\gamma^t\Big(r(a_t,s_t) - \alpha\log\pi(\cdot|s_t)\Big)\Big] \tag{15}$$

is $\pi^* = \frac{\exp(\frac{1}{\alpha}Q(s,a))}{Z}$. Here, $Q(s,a)$ is the soft Q-function defined as

$$
\begin{aligned}
Q(s,a) &= \mathbb{E}_{(s_t,a_t)\sim\rho_\pi}\Big[\sum_t \gamma^t\Big(r(a_t,s_t) - \alpha\log p(\pi(\cdot|s_t))\Big)|s_0 = s, a_0 = a\Big] \\
&= r(a,s) + \alpha\mathcal{H}(\pi(\cdot|s)) + \mathbb{E}_{\pi(a'|s)\rho_\pi(s')}\Big[Q(s',a')\Big].
\end{aligned}
\tag{16}
$$

Consequently, we deduce that $\pi^*$ is also the solution of the expected KL divergence:

$$\pi^* = \arg\min_\pi \sum_t \mathbb{E}_{s_t\sim\rho_\pi}\Big[D_{KL}\big(\pi(\cdot|s_t)\|\exp(Q(s_t,\cdot)/\alpha)/Z\big)\Big]. \tag{17}$$

*Proof.* We express the MaxEnt loss as a function of $Q(s,a)$, *i.e.*, $J(\pi) = \mathbb{E}_{(s,a)\sim\rho_\pi}\Big[Q(s,a)\Big]$. To find $\pi^* = \arg\max_\pi J(\pi)$ under the constraint $\int_a \pi(a|s)da = 1$, we evaluate the Lagrangian (with $\lambda \in \mathbb{R}$ being the Lagrange multiplier):

$$\mathcal{L}(\pi,\lambda) = \mathbb{E}_{(s,a)\sim\rho_\pi}\Big[Q(s,a)\Big] + \lambda\Big(\int_a \pi(a|s)da - 1\Big), \tag{18}$$

and compute $\frac{\partial\mathcal{L}(\pi,\lambda)}{\partial\pi(a|s)}$:

$$
\begin{aligned}
\frac{\partial\mathcal{L}(\pi,\lambda)}{\partial\pi(a|s)} &= \frac{\partial}{\partial\pi(a|s)}\Big(\int_s\int_a \pi(a|s)\rho_\pi(s)Q(s,a)da\,ds + \lambda\Big(\int_a \pi(a|s)da - 1\Big)\Big) \\
&= \frac{\partial}{\partial\pi(a|s)}\Big(\pi(a|s)\rho_\pi(s)\Big(r(a,s) - \alpha\log\pi(a|s) + \mathbb{E}_{\pi(a'|s)\rho_\pi(s')}\Big[Q(s',a')\Big]\Big)\Big) + \lambda \\
&= \rho_\pi(s)\Big(r(a,s) + \mathbb{E}_{\pi(a'|s)\rho_\pi(s')}[Q(s',a')]\Big) - \alpha\rho_\pi(s)\frac{\partial}{\partial\pi(a|s)}\Big(\pi(a|s)\log\pi(a|s)\Big) + \lambda \\
&= \rho_\pi(s)\Big(r(a,s) + \mathbb{E}_{\pi(a'|s)\rho_\pi(s')}[Q(s',a')]\Big) - \alpha\rho_\pi(s)\Big(\log\pi(a|s) + 1\Big) + \lambda.
\end{aligned}
$$

Setting $\frac{\partial\mathcal{L}(\pi,\lambda)}{\partial\pi(a|s)}$ to 0:

$$
\begin{aligned}
\frac{\partial\mathcal{L}(\pi,\lambda)}{\partial\pi(a|s)} = 0 \iff & \Big(r(a,s) + \mathbb{E}_{\pi(a'|s)\rho_\pi(s')}[Q^\pi(s',a')]\Big) - \alpha + \frac{\lambda}{\rho_\pi(s)} = \alpha\log\pi(a|s) \\
\iff & \frac{1}{\alpha}\Big(r(a,s) + \mathbb{E}_{\pi(a'|s)\rho_\pi(s')}[Q(s',a')]\Big) - 1 + \frac{\lambda}{\alpha\rho_\pi(s)} = \log\pi(a|s) \\
\iff & \pi(a|s) = \frac{\exp\Big(\frac{1}{\alpha}\Big(r(a,s) + \mathbb{E}_{\pi(a'|s)\rho_\pi(s')}[Q(s',a')]\Big)\Big)}{\exp\Big(1 - \frac{\lambda}{\alpha\rho_\pi(s)}\Big)} \\
\iff & \pi(a|s) = \frac{\exp\Big(\frac{1}{\alpha}\Big(r(a,s) + \mathcal{H}(\pi(\cdot|s)) + \mathbb{E}_{\pi(a'|s)\rho_\pi(s')}[Q(s',a')]\Big)\Big)}{\exp\Big(1 - \frac{\lambda}{\alpha\rho_\pi(s)}\Big)} \\
\iff & \pi(a|s) = \frac{\exp\Big(\frac{1}{\alpha}Q(s,a)\Big)}{\exp\Big(\frac{\mathcal{H}(\pi(\cdot|s))}{\alpha} + 1 - \frac{\lambda}{\alpha\rho_\pi(s)}\Big)}
\end{aligned}
\tag{19}
$$

We choose $\lambda$ such that $\int_a \pi(a|s)da = 1$, *i.e.*,

$$\int_a \frac{\exp\left(\frac{1}{\alpha}Q(s,a)\right)}{\exp\left(\frac{\mathcal{H}(\pi(\cdot|s))}{\alpha}+1-\frac{\lambda}{\alpha\rho_\pi(s)}\right)}da = 1 \iff \lambda = -\alpha\rho_\pi(s)\left(\log\int_a \exp\left(\frac{1}{\alpha}Q(s,a)\right)da - \frac{\mathcal{H}(\pi(\cdot|s))}{\alpha}-1\right).$$

(20)

Hence, $\pi^*(a|s) = \exp(\frac{1}{\alpha}Q(s,a))/\int_a \exp(\frac{1}{\alpha}Q(s,a))$. A similar proof follows for any state and action pairs. Trivially, $\pi^*$ is also the global minimum of Eq.(17). $\square$

## E  DERIVATION OF THE ACTOR OBJECTIVE (EQ.(9))

In the following, we prove that the objective

$$\arg\min_\theta \mathbb{E}_{s_t\sim\mathcal{D}}\left[D_{KL}\left(\pi_\theta(\cdot|s_t)\middle\|\exp\left(\frac{1}{\alpha}Q_\phi(s_t,\cdot)\right)/Z(\phi)\right)\right]$$

is equivalent to

$$\arg\max_\theta \mathbb{E}_{s_t\sim\mathcal{D},a_t\sim\pi_\theta(a_t|s_t)}\left[Q_\phi(s_t,a_t)\right] + \mathbb{E}_{s_t}\left[\alpha\mathcal{H}(\pi_\theta(a_t|s_t))\right],$$

with $\mathcal{D}$ being a replay buffer.

*Proof.*

$$
\begin{aligned}
\theta^* &= \arg\min_\theta \mathbb{E}_{s_t\sim\mathcal{D}}\left[D_{KL}\left(\pi_\theta(\cdot|s_t)\middle\|\exp\left(\frac{1}{\alpha}Q_\phi(s_t,\cdot)\right)/Z(\phi)\right)\right]\\
&= \arg\min_\theta \mathbb{E}_{s_t\sim\mathcal{D},a_t\sim\pi_\theta(a_t|s_t)}\left[\log(\pi_\theta(a_t|s_t)) - \left(\frac{1}{\alpha}Q_\phi(s_t,a_t) - \log Z(\phi)\right)\right]\\
&= \arg\min_\theta \mathbb{E}_{s_t\sim\mathcal{D},a_t\sim\pi_\theta(a_t|s_t)}\left[\log(\pi_\theta(a_t|s_t)) - \frac{1}{\alpha}Q_\phi(s_t,a_t)\right]\\
&= \arg\max_\theta \mathbb{E}_{s_t\sim\mathcal{D},a_t\sim\pi_\theta(a_t|s_t)}\left[-\log(\pi_\theta(a_t|s_t)) + \frac{1}{\alpha}Q_\phi(s_t,a_t)\right]\\
&= \arg\max_\theta \mathbb{E}_{s_t\sim\mathcal{D},a_t\sim\pi_\theta(a_t|s_t)}\left[\frac{1}{\alpha}Q_\phi(s_t,a_t) + \mathcal{H}(\pi_\theta(a_t|s_t))\right]\\
&= \arg\max_\theta \mathbb{E}_{s_t\sim\mathcal{D},a_t\sim\pi_\theta(a_t|s_t)}\left[Q_\phi(s_t,a_t)\right] + \mathbb{E}_{s_t\sim\mathcal{D}}\left[\alpha\mathcal{H}(\pi_\theta(a_t|s_t))\right]
\end{aligned}
$$

$\square$

## F  PROOF OF THEOREM 3.1

**Theorem.** *Let $F : \mathbb{R}^n \to \mathbb{R}^n$ be an invertible transformation of the form $F(a) = a + \epsilon h(a)$. We denote by $q^L(a^L)$ the distribution obtained from repeatedly ($L$ times) applying $F$ to a set of action samples (called "particles") $\{a^0\}$ from an initial distribution $q^0(a^0)$, i.e., $a^L = F \circ F \circ \cdots \circ F(a^0)$. Under the condition $\epsilon||\nabla_{a_i} h(a_i)||_\infty \ll 1$, the closed-form expression of $\log q^L(a^L)$ is:*

$$\log q^L(a^L) = \log q^0(a^0) - \epsilon \sum_{l=0}^{L-1} \text{Tr}(\nabla_{a^l} h(a^l)). \tag{21}$$

*Proof.* Based on the change of variable formula (Appendix C.2), when for every iteration $l \in [1, L]$, the transformation $a^l = L(a^{l-1})$ (of the action sampler in our paper) is invertible, we have:

$$q^l(a^l) = q^{l-1}(a^{l-1}) \left| \det \frac{da^l}{da^{l-1}} \right|^{-1}, \forall l \in [1, L].$$

By induction, we derive the probability distribution of sample $a^L$:

$$q^L(a^L) = q^0(a^0) \prod_{l=1}^{L} \left| \det \frac{da^l}{da^{l-1}} \right|^{-1} = q^0(a^0) \prod_{l=0}^{L-1} \left| \det \left( I + \epsilon \nabla_{a^l} h(a^l) \right) \right|^{-1}$$

By taking the $\log$ for both sides, we obtain:

$$\log q^L(a^L) = \log q^0(a^0) - \sum_{l=0}^{L-1} \log \left| \det \left( I + \epsilon \nabla_{a^l} h(a^l) \right) \right|.$$

Let $A = I + \epsilon \nabla_{a^l} h(a^l)$, under the assumption $\epsilon||\nabla_{a_i} h(a_i)||_\infty \ll 1$, *i.e.*, $||A - I||_\infty \ll 1$, we apply the corollary of Jacobi's formula (Appendix C.3) and get

$$\log q^L(a^L) \approx \log q^0(a^0) - \sum_{l=0}^{L-1} \text{Tr} \left( (I + \epsilon \nabla_{a^l} h(a^l)) - I) \right) + \mathcal{O}(\epsilon^2 dL)$$

$$\approx \log q^0(a^0) - \epsilon \sum_{l=0}^{L-1} \text{Tr} \left( \nabla_{a^l} h(a^l) \right) + \mathcal{O}(\epsilon^2 dL).$$

Here, $d$ is the action space dimension. □

# G    SAMPLERS INVERTIBILITY PROOFS

We start by state the implicit function theorem which we will be using in the following proofs.

**Theorem G.1** (**Implicit function theorem**). *Let $f : \mathbb{R}^n \to \mathbb{R}^n$ be continuously differentiable on some open set containing $a$, and suppose $\det(Jf(a)) = \det(\nabla_a f(a)) \neq 0$. Then, there is some open set $V$ containing $a$ and an open $W$ containing $f(a)$ such that $f : V \to W$ has a continuous inverse $f^{-1} : W \to V$ which is differentiable $\forall y \in W$.*

## G.1    STOCHASTIC GRADIENT LANGEVIN DYNAMICS

**Proposition** (SGLD). *The SGLD update in Eq.(12) is not invertible.*

*Proof.* We show that SGLD are not invertible using two different methods: (1) We show that SGLD is not a bijective transformation, (2) Using the implicit function theorem, we show that the Jacobian of the dynamics is not invertible.

### G.1.1    METHOD1: SGLD IS NOT INVERTIBLE $\iff$ SGLD IS NOT A BIJECTION

The update rule $F(\cdot)$ for SGLD and DGLD are given by Eq.(12) and Eq.(13), respectively. In the following, we drop the dependency on the time step for ease of notation.

**Injectivity** is equivalent to checking: $F(a_1) = F(a_2) \implies a_1 = a_2$. This, however, doesn't hold in case of SGLD as the noise terms $\xi_1$ and $\xi_2$ can be chosen such that the equality

$$a_1 + \epsilon \nabla_{a_1} \log p(a_1) + \sqrt{2\epsilon}\xi_1 = a_2 + \epsilon \nabla_{a_2} \log p(a_2) + \sqrt{2\epsilon}\xi_2$$

holds with $a_1 \neq a_2$. Therefore, SGLD is not injective. The same holds for DGLD, where the equality $a_1 + \nabla_{a_1} \log p(a_1) = a_2 + \nabla_{a_2} \log p(a_2)$ can be valid for $a_1 \neq a_2$. A counter-example: $a_1 = a_2 + \eta$ and $\eta + \nabla_{a_1} \log p(a_1) = \nabla_{a_2} \log p(a_2)$, with $\eta$ being an arbitrary constant.

**Surjectivity** is equivalent to checking: $\forall a^{l+1} \in \mathbb{R}^d, \exists a^l \in \mathbb{R}^d$ s.t. $a^{l+1} = F(a^l)$. Assume that $a^{l+1} = a^l + \epsilon \nabla_{a^l} \log p(a^l)$, and $a^l \in \mathbb{R}^d$, we can always choose an adaptive learning rate $\epsilon$ such that $\epsilon \nabla_{a^l} \log p(a^l) = a^{l+1} - a^l$.

### G.1.2    METHOD2: IMPLICIT FUNCTION THEOREM

We compute the derivative of the update rule in Eq 12 with respect to $a$: $J_F = I + \epsilon \nabla_a^2 \log p(a)$. It's possible for $J_F$ not to be invertible, *e.g.*, in case $I = -\epsilon \nabla_a^2 \log p(a)$. Hence, in general $F(a)$ is not guaranteed to be a bijection. $\qquad\square$

## G.2    HAMILTONIAN MONTE CARLO (HMC)

**Proposition** (HMC). *The HMC update in Eq.(14) is not invertible w.r.t. $a$.*

Neal et al. (2011) show that HMC update rule is only invertible with respect to the $(a, v)$, *i.e.*, when conditioning on $v$. Since $v$ is sampled from a random distribution, it has the effect of the noise variable in SGLD. Hence, a similar proof applies.

## G.3    STEIN VARIATIONAL GRADIENT DESCENT

**Proposition** (SVGD). *Under the assumption that $\epsilon \ll \sigma$, the update rule of SVGD dynamics defined in Eq.(1) with an RBF kernel is invertible.*

### G.3.1    METHOD2: SVGD IS INVERTIBLE $\iff$ SVGD IS A BIJECTION

**Injectivity.** The equality $F(a_1) = F(a_2)$:

$$a_1 + \frac{\epsilon}{m}\sum_{j=1}^{m}\left[k(a_j, a_1)\nabla_{a_j}\log p(a_j) - \nabla_{a_j}k(a_j, a_1)\right] = a_2 + \frac{\epsilon}{m}\sum_{l=1}^{m}\left[k(a_l, a_2)\nabla_{a_l}g(a_l) - \nabla_{a_l}k(a_l, a_2)\right]$$

is too complex to hold for a solution other than $a_1 = a_2$ given the sum over multiple particles on both sides and the dependency on the kernel.

$\implies$ not obvious ( depends on the Kernel)

**Surjectivity.** Similarly, to Langevin dynamics, surjectivity can be achieved by choosing a suitable learning rate.

### G.3.2 METHOD 2: IMPLICIT FUNCTION THEOREM

We start by proving the proposition above for the 1-Dimensional case, *i.e.*, $a \in \mathbb{R}$. Then, we extend the proof to the multi-dimensional case, *i.e.*, $a \in \mathbb{R}^d$.

**1-Dimensional Case.** We prove that $F$ is invertible by showing that $F$ is bijective, which is equivalent to showing that $F$ is strictly monotonic, *i.e.*, $\nabla_{a_i} F(a_i) > 0$ or $\nabla_{a_i} F(a_i) < 0, \quad \forall a_i$.
Computing the derivative of the SVGD update (Eq. 1) rule w.r.t $a_i$ results in:

$$\nabla_{a_i} F(a_i) = 1 + \frac{\varepsilon}{m} \sum_{i=1}^{m} \nabla_{a_i} k(a_i, a_j) \nabla_{a_j} g(a_j) + \nabla_{a_i} \nabla_{a_j} k(a_i, a_j).$$

For $k(a_i, a_j) = e^{-\frac{\|a_i - a_j\|^2}{2\sigma^2}}$, we have:
$$\begin{cases} \nabla_{a_j} k(a_i, a_j) = \frac{-2(a_i - a_j)}{2\sigma^2} k(a_i, a_j) = \frac{-(a_i - a_j)}{\sigma^2} k(a_i, a_j) \\ \nabla_{a_j} k(a_i, a_j) = \frac{(a_i - a_j)}{\sigma^2} k(a_i, a_j) \\ \nabla_{a_i} \nabla_{a_j} = \frac{1}{\sigma^2} k(a_i, a_j) \left(1 - \frac{1}{\sigma^2} \|a_i - a_j\|^2\right) \end{cases}$$

Hence,

$$\nabla_{a_i} F(a_i) = 1 + \frac{\epsilon}{m} \sum_{i=1}^{m} \frac{k(a_i, a_j)}{\sigma^2} \left(-(a_i - a_j)\nabla_{a_j} \log p(a_j) + 1 - \frac{\|a_i - a_j\|}{\sigma^2}\right).$$

Next, under the condition $\epsilon < \sigma$, we show that $\nabla_{a_i} F(a_i) > 0, \forall a_i$. This is equivalent to showing that $\nabla_{a_i} F(a_i) > -1$.

$$\nabla_{a_i} F(a_i) > -1 \iff \frac{\epsilon}{m\sigma^2} \sum_{j=1}^{m} k(a_i, a_j) \left(-(a_i - a_j)\nabla_{a_j} \log p_{a_j}(a_j) + 1 - \frac{\|a_i - a_j\|^2}{\sigma^2}\right) > -1$$

We compute a lower bound on the LHS and investigate when it's strictly larger than $-1$. We can safely assume that $-3\sigma \leq k(a_i, a_j)(a_i - a_j) \leq 3\sigma$ and $-3\sigma \leq k(a_i, a_j)\|a_i - a_j\|^2 \leq 3\sigma$. We compute the lower bound as: $\sum_{j=1}^{m} \frac{\epsilon\alpha}{m\sigma^2} \left(-3\sigma\|\nabla_{x_j} \log p(x_j)\| + 1 - \frac{(3\sigma)^2}{\sigma^2}\right) = \frac{\epsilon\alpha}{m\sigma^2} \left(-3\sigma \sum_{j=1}^{m} \|\nabla_{x_j} \log p(x_j)\| - 8m\right)$. This results in:

$$\nabla_{a_i} F(a_i) > \frac{\epsilon\alpha}{m\sigma^2} \left(-3\sigma \sum_{j=1}^{m} \|\nabla_{a_j} \log p(a_j)\| - 8m\right) > -1 \iff \sum_{j=1}^{m} \|\nabla_{a_j} \log p(a_j)\| < \frac{m\sigma}{3\epsilon\alpha} - \frac{8m}{3\sigma}$$

(22)

Hence, $\max_{a_j} \|\nabla_{a_j} \log p(a_j)\| < \frac{\sigma}{3\epsilon\alpha} - \frac{8}{3\sigma}$. The LHS is guaranteed to be a large positive number when $\epsilon \ll \sigma$.

**Multi-Dimensional Case.** We assume that $\log p(a_j)$ is continuously differentiable. Note that in practice, this can be easily satisfied by choosing the activation function to be Elu instead of Relu. We can easily show that:

$$\nabla_{a_i} F(a_i) = I + \frac{\epsilon}{m\sigma^2} \sum_{i=1}^{m} k(a_i, a_j) \left(-\nabla_{a_j} \log p(a_j)(a_i - a_j)^\top - \frac{1}{\sigma^2}(a_i - a_j)(a_i - a_j)^\top + I\right)$$

Next, we will show that $\nabla_{a_i} F(a_i)$ is diagonally dominated and is, hence, invertible, *i.e.*, $\det(\nabla_{a_i} h(a_i)) \neq 0$. For this, we show that $\nabla_{a_i} h(a_i)|_{kl} < 1, \forall k, l \in [1, d]$.

$$\nabla_{a_i} h(a_i)|_{kl} = \frac{1}{m} \sum_{i=1}^{m} k(a_i, a_j) \left(-\partial_{a_j^{(k)}} \log p(a_j)(a_i^{(l)} - a_j^{(l)}) - (a_i^{(k)} - a_j^{(k)})(a_i^{(l)} - a_j^{(l)}) + 1\right)$$

Following the proof in Section G.3.2 for the 1-Dimensional case, we show that $\nabla_{a_i} h(a_i)|_{kl} \ll 1$ if $\sigma \ll \epsilon$.

## H  DERIVATION OF CLOSED-FORM LIKELIHOOD FOR SAMPLERS

### H.1  PROOF OF THEOREM 3.3

**Theorem.** *The closed-form estimate of the log-likelihood* $\log q^L(a^L|s)$ *for the SVGD-based sampler with an RBF kernel* $k(\cdot,\cdot)$ *is*

$$\log q^L(a^L|s) \approx \log q^0(a^0|s) - \frac{\epsilon}{m\sigma^2} \sum_{l=0}^{L-1} \sum_{\substack{j=1 \\ a^l \neq a_j^l}}^{m} k(a_j^l, a^l) \left( -(a^l - a_j^l)^\top \nabla_{a_j^l} Q(s, a_j^l) - \frac{\alpha}{\sigma^2} \|a^l - a_j^l\|^2 + d\alpha \right),$$

*where* $d$ *is the feature space dimension.*

*Proof.* We generate a chain of samples using SVGD starting from $a^0 \sim q^0$, and following the update rule $a_i^{l+1} \leftarrow a_i^l + \epsilon\, h(a_i^l, s)$, where $h(a_i^l, s) = \mathbb{E}_{a_j^l \sim q^l}\left[ k(a_i^l, a_j^l) \nabla_{a_j^l} Q(s, a_j^l) + \nabla_{a_j^l} k(a_i^l, a_j^l) \right]$ and $k(a_i^l, a_j^l) = \exp\left(-\frac{\|a_i^l - a_j^l\|^2}{2\sigma^2}\right)$. This update rule is the optimal direction in the reproducing kernel Hilbert space of $k(\cdot, \cdot)$ for minimizing the KL divergence objective (actor loss):

$$\pi^* = \arg\min_\pi \sum_t \mathbb{E}_{s_t \sim \rho_\pi}\left[ D_{KL}\left( \pi(\cdot|s_t) \| \exp(Q(s_t, \cdot)/\alpha)/Z \right) \right]. \tag{23}$$

According to Proposition 3.2, the iteration step (Eq.(1)) is invertible. Hence, following Theorem 3.1 and substituting $h(a_i^l, s)$ with the above formula for SVGD, for each action particle $a_i^L$ we obtain:

$$\log q^L(a_i^L) \approx \log q^0(a_i^0) - \frac{1}{m} \sum_{l=0}^{L-1} \sum_{\substack{j=1 \\ a_i^l \neq a_j^l}}^{m} \Big[ \underbrace{\mathrm{Tr}\left( \nabla_{a_i^l}(k(a_i^l, a_j^l) \nabla_{a_j^l} Q(s, a_j^l)) \right)}_{\textcircled{1}} + \underbrace{\mathrm{Tr}\,\alpha\left( \nabla_{a_i^l} \nabla_{a_j^l} k(a_i^l, a_j^l) \right)}_{\textcircled{2}} \Big].$$

Note that we empirically approximate the expectation in $h(a_i^l, s)$ by an empirical mean over particles that are different from $a_i^l$, in order to avoid computing Hessians in the derivation below. Next we compute simplifications for terms ① and ② respectively. In the following, we denote by $(\cdot)^{(k)}$ the $k$-th dimension of the vector.

**Term ①:**

$$
\begin{aligned}
\mathrm{Tr}\left( \nabla_{a_i^l}(k(a_j^l, a_j^l) \nabla_{a_j^l} Q(s, a_j^l)) \right) &= \mathrm{Tr}\left( \nabla_{a_i^l} k(a_j^l, a_j^l)(\nabla_{a_j^l} Q(s, a_j^l))^\top + k(a_j^l, a_j^l) \nabla_{a_i^l} \nabla_{a_j^l} Q(s, a_j^l) \right) \\
&= \sum_{t=1}^d \frac{\partial k(a_j^l, a_j^l)}{\partial (a_i^l)^{(t)}} \frac{\partial Q(s, a_j^l)}{\partial (a_i^l)^{(t)}} + 0 \\
&= (\nabla_{a_i^l} k(a_j^l, a_j^l))^\top \nabla_{a_j^l} Q(s, a_j^l) \\
&= -\frac{\alpha}{\sigma^2} k(a_j^l, a_j^l)(a_i^l - a_j^l)^\top \nabla_{a_j^l} Q(s, a_j^l)
\end{aligned}
$$

**Term ②:**

$$
\begin{aligned}
\mathrm{Tr}\left( \nabla_{a_i^l} \nabla_{a_j^l} k(a_i^l, a_j^l) \right) &= \alpha\, \mathrm{Tr}\left( \nabla_{a_i^l}\left( \frac{1}{\sigma^2} k(a_i^l, a_j^l)(a_i^l - a_j^l) \right) \right) \\
&= \frac{\alpha}{\sigma^2} \mathrm{Tr}\left( \nabla_{a_i^l} k(a_i^l, a_j^l)(a_i^l - a_j^l)^\top + k(a_i^l, a_j^l) \cdot I \right) \\
&= \frac{\alpha}{\sigma^2} \mathrm{Tr}\left( -\frac{1}{\sigma^2} k(a_i^l, a_j^l)(a_i^l - a_j^l)(a_i^l - a_j^l)^\top + k(a_i^l, a_j^l) \cdot I \right) \\
&= \frac{\alpha}{\sigma^2} \sum_{t=1}^d \left( -\frac{1}{\sigma^2} k(a_i^l, a_j^l)(a_i^l - a_j^l)^{(t)}(a_i^l - a_j^l)^{(t)} + k(a_i^l, a_j^l) \right) \\
&= -\frac{\alpha}{\sigma^4} \times k(a_i^l, a_j^l) \|a_i^l - a_j^l\|^2 + \frac{\alpha}{\sigma^2} \times d \times k(a_i^l, a_j^l) \\
&= k(a_i^l, a_j^l)\left( -\frac{\alpha}{\sigma^4} \|a_i^l - a_j^l\|^2 + \frac{d\alpha}{\sigma^2} \right)
\end{aligned}
$$

By combining **Terms** ① and ②, we obtain:

$$\log q^L(a_i^L) \approx \log p^0(a_i^0) - \frac{\epsilon}{m\sigma^2} \sum_{l=0}^{L-1} \sum_{j=1}^{m} k(a_j^l, a_j^l) \left( -(a_i^l - a_j^l)^\top \nabla_{a_j^l} Q(s, a_j^l) - \frac{\alpha}{\sigma^2} \|a_i^l - a_j^l\|^2 + d\alpha \right)$$

Proof done if we take a generic action particle $a_i$ in place of $a$. $\qquad\square$

# I ADDITIONAL RESULTS: ENTROPY EVALUATION

The SVGD hyperparameters for this set of experiments are summarized in Table 2. We include additional figures for (1) the effect of (2) the kernel variance (Figure 9) and (2) number of SVGD steps and particles (Figure 10).

Table 2: Parameters

|  | Parameter | Value |
|---|---|---|
| Figure 4a-4b | Target distribution
Initial distribution | $p = \mathcal{N}([-0.69, 0.8], [[1.13, 0.82], [0.82, 3.39]])$
$q^0 = \mathcal{N}([0, 0], 6\boldsymbol{I})$ |
| Figure 4c | Target distribution
Initial distribution | $p_{\mathrm{GMM}_M} = \sum_{i=1}^{M} \mathcal{N}([0, 0], 0.1\boldsymbol{I})/M$
$q^0 = \mathcal{N}([0, 0], 6\boldsymbol{I})$ |
| Default
SVGD
parameters | Learning rate
Number of steps
Number of particles
Kernel variance | $\epsilon = 0.5$
$L = 200$
$m = 200$
$\sigma = 5$ |

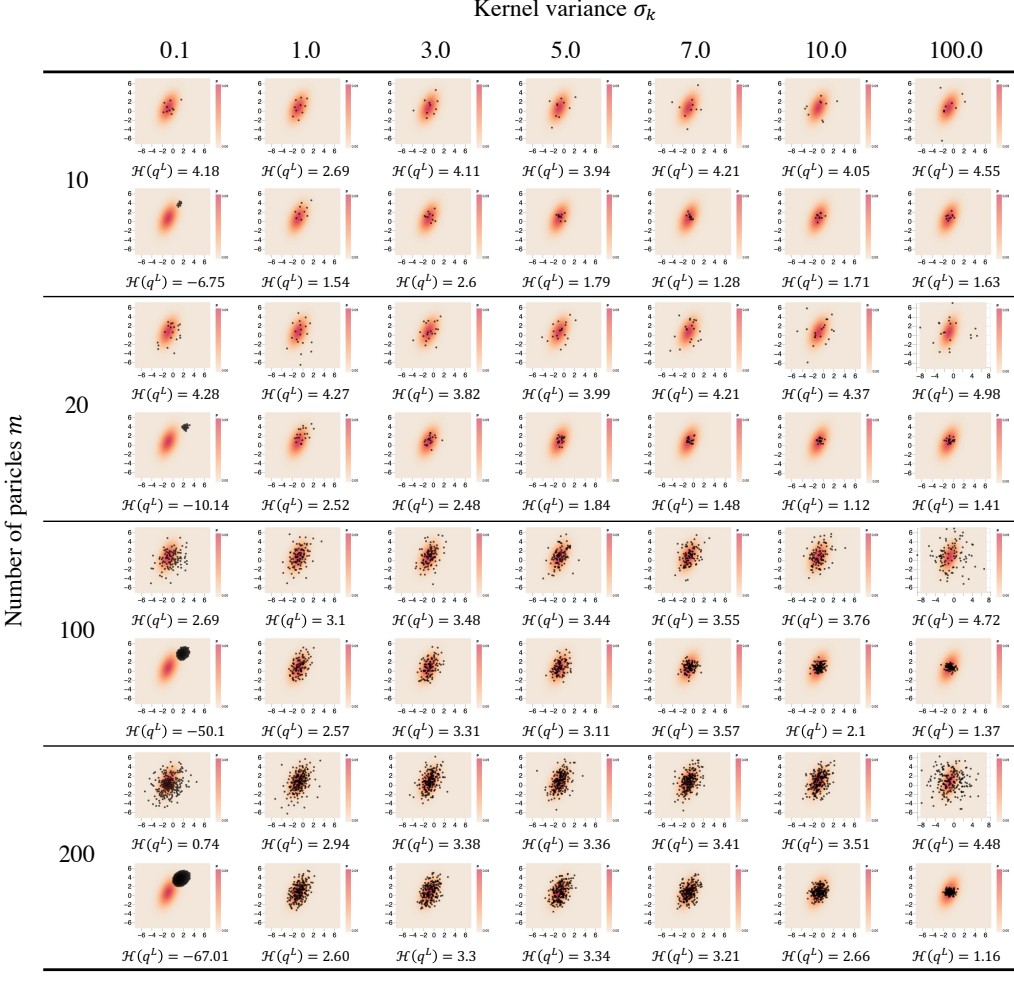

Figure 9: Visualization of the particles after $L$ steps of SVGD for the different configurations of kernel variance $\sigma$ and number of particles $m$ in Figure 4b.

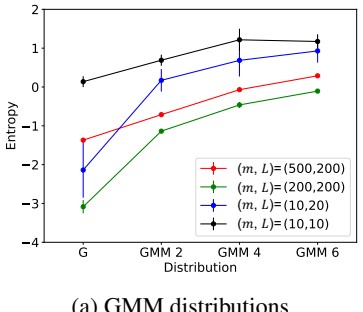

(a) GMM distributions

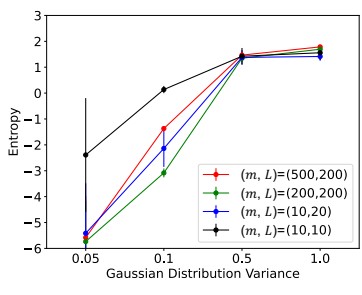

(b) Gaussian distributions

Figure 10: Sensitivity of our entropy formula to the number of SVGD steps ($L$) and particles ($m$). Our entropy consistently increases with increasing $\sigma$ and increasing number of GMM components, even when a small number of SVGD steps and particles is used *e.g.*, $L = 10, m = 10$.

## J   ADDITIONAL RESULTS: MULTI-GOAL RESULTS

Hyperparameters are reported in Table 3. Additionally, we include results for (1) the effect of the parametrization of the initial distribution (Figure 14), (2) the entropy heatmap (Figure 13), (3) the effect of the entropy on the learned Q-landscapes (Figure 14 and Figure 15), (4) the robustness/adaptability of the learned policies (Figure 16) and (5) Amortized $S^2$AC results (Figure 17).

Table 3: Hyperparameters for multi-goal environment.

| | Hyperparameter | Value |
|---|---|---|
| Training | Optimizer | Adam |
| | Learning rate | $3 \cdot 10^{-4}$ |
| | Batch size | 100 |
| Deepnet | Number of hidden layers (all networks) | 2 |
| | Number of hidden units per layer | 256 |
| | Nonlinearity | ReLU |
| RL | Discount factor $\gamma$ | 0.8 |
| | Replay buffer size $|\mathcal{D}|$ | $10^6$ |
| | Target smoothing coefficient | 0.005 |
| | Target update interval | 1 |
| SVGD | initial distribution $q^0$ | $\mathcal{N}(\mathbf{0}, 0.3\boldsymbol{I})$ |
| | Learning rate $\epsilon$ | 0.01 |
| | Number of steps $L$ | 10 |
| | Number of particles $m$ | 10 |
| | Particles range (num. std) $t$ | 3 |
| | Kernel variance | $\sigma = \dfrac{\sum_{i,j} \|a_i - a_j\|^2}{4(2 \log m + 1)}$ |

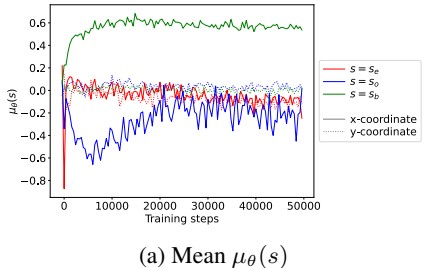

(a) Mean $\mu_\theta(s)$

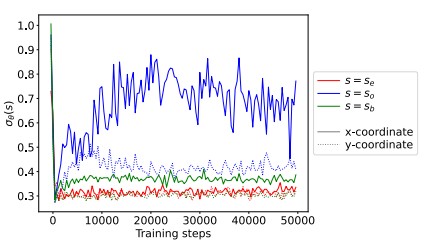

(b) Standard deviation $\sigma_\theta(s)$

Figure 11: Trends of $x$ and $y$ coordinates for the mean and standard deviation of the parameterized initial distribution for some critical states, during training.

**Distribution of reached goals for the multi-goal environment.** Figure 12 shows the distribution of reached goals for S$^2$AC/SAC for the agents in Figure 6. Trajectories are collected from 20 episodes of 20 different agents trained with 20 different seeds for each algorithm. We observe that with higher $\alpha$'s, more agent trajectories converge to the left two goals (G2 and G3), which is not the case for SAC and SQL. This shows that S$^2$AC learns a more optimal solution to the MaxEnt objective in Eq.(2).

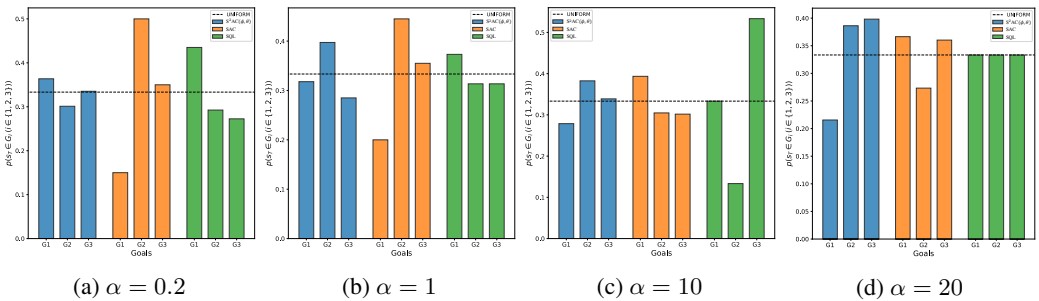

(a) $\alpha = 0.2$     (b) $\alpha = 1$     (c) $\alpha = 10$     (d) $\alpha = 20$

Figure 12: Distribution of reached goals for S$^2$AC, SAC and SQL with different $\alpha$'s. The x-axis denotes different goals. The y-axis represents the ratio of trajectories that reach the goal.

**Entropy heatmap of S$^2$AC in the multi-goal environment.** Figure 13 shows the entropy heatmap of S$^2$AC with different $\alpha$'s. A brighter color corresponds to higher entropy. For S$^2$AC, the higher $\alpha$, the higher the entropy on the left quadrant compared to the right one, i.e., the more contrast between the left and the right quadrants. For instance, In Figure 13d (S$^2$AC, $\alpha = 20$), notice a clear green/yellow patch spanning the left side, while the right side is mostly dark blue except for the edges.

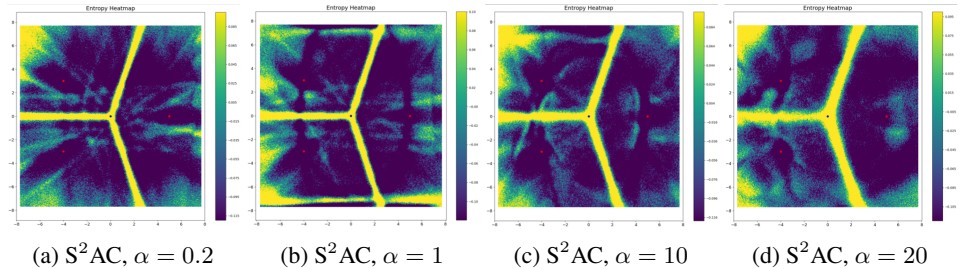

(a) S$^2$AC, $\alpha = 0.2$    (b) S$^2$AC, $\alpha = 1$    (c) S$^2$AC, $\alpha = 10$    (d) S$^2$AC, $\alpha = 20$

Figure 13: The entropy heatmap of S$^2$AC in the multi-goal environment for different $\alpha$

**Smoothness of the Q-landscapes.** To assess the effect of the entropy, we visualize the Q-landscapes corresponding to six typical states $s \in \{s_o, s_a, s_b, s_c, s_d, s_e\}$ (marked in blue on the upper left of Figure 14) across different trajectories to the goal and report their associated entropy $\mathcal{H}(\cdot|s)$ (bottom left of Figure 14). The blue dots correspond to 10 SVGD particles at convergence. We observe that the Q-landscape becomes smoother with increasing $\alpha$. For instance, notice how the modes for state $s_c$ become more connected. Quantified measurements of smoothness are in Figure 15. We use two metrics $M_1$ and $M_2$ to measure the smoothness of the learned Q-landscape: (1) $M_1$: the average over the L1-norm of the gradient of the Q-value with respect to the actions across trajectories, i.e., $\mathbb{E}_{\tau \sim \pi(a|s)}\left[\mathbb{E}_{(s_t,a_t)\in\tau}\left[\frac{||\nabla_{a_t}Q(s_t,a_t)||_1}{d}\right]\right]$. (2) $M_2$: The average over the L1-norm of the Hessian of the Q-value with respect to the actions across trajectories, i.e., $\mathbb{E}_{\tau \sim \pi(a|s)}\left[\mathbb{E}_{(s_t,a_t)\in\tau}\left[\frac{1}{d^2}\sum_{i,j}|\nabla_{a_t}^2 Q(s_t,a_t)|_{i,j}\right]\right]$. Figure 15 shows that increasing $\alpha$ leads to consistently smaller gradients (Figure 15a) and less curvature (Figure 15b). Hence, the entropy results in a smoother landscape that helps the sampling convergence.

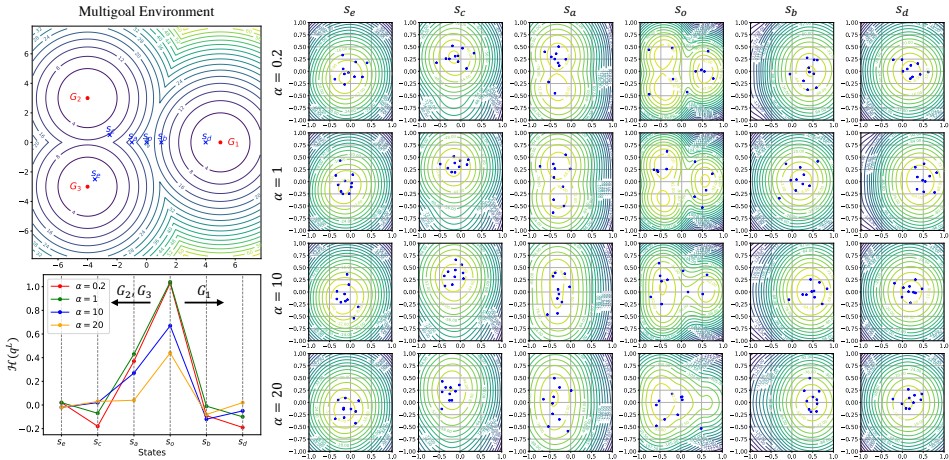

Figure 14: Results on the multi-goal environment. Increasing $\alpha$ yields smoother landscapes (*e.g.*, $s_o$). Notice how the modes become more connected (*e.g.*, for $s = s_a$ with increasing $\alpha$). The entropy at the different states is reported in the lower left figure.

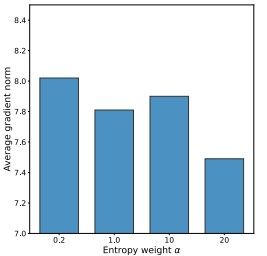

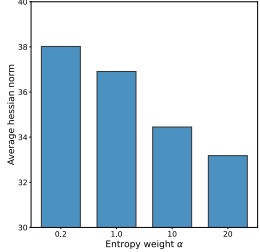

(a) Average gradient across trajectories

(b) Average Hessian across trajectories

Figure 15: Quantitative evaluation of the smoothness the Q-landscape of S$^2$AC for different $\alpha$'s.

**Parametrization of** $q^0$. In Figure 11, we visualizes the coordinates of the mean $\mu_\theta(s)$ and standard deviation $\sigma_\theta(s)$ of $q^0_\theta$ at different states $s \in \{s_o, s_e, s_b\}$ in the multigoal environment. As training goes on, $\mu_\theta(s)$ shifts closer to the nearest goals. For example, $\mu_\theta(s_b)$ becomes more positive during the training as it is shifting to $G_1$. Additionally, the model learns a high variance $\sigma_\theta(s)$ for the multimodal state $s_o$ and becomes more deterministic for the unimodal ones (*e.g.*, $s_e$ and $s_b$). As a result, in Figure 7, we observe that S$^2$AC($\phi, \theta$) requires a smaller number of steps to convergence than S$^2$AC($\phi$).

**Entropy estimation.** Figure 14 shows that the entropy is higher for states on the left side due to the presence of two goals, as opposed to a single goal on the right side (*e.g.*, $\mathcal{H}(\pi_\theta(\cdot|s_a)) < \mathcal{H}(\pi_\theta(\cdot|s_o))$). Also, the entropy decreases when approaching the goals (*e.g.*, $\mathcal{H}(\pi_\theta(\cdot|s_d)) < \mathcal{H}(\pi_\theta(\cdot|s_b)) < \mathcal{H}(\pi_\theta(\cdot|s_o))$). The same is valid along the paths to goal $G_1$.

**Robustness/Adaptability.** In Figure 16, we report the distribution of reached goals after hitting an obstacle for S$^2$AC, SAC and SQL for different $\alpha$'s. Notice that S$^2$AC robustness, measured by the probability of reaching the goal for S$^2$AC is consistently increasing with increasing $\alpha$. Intuitively, exploration is better with large values of $\alpha$, leading to better learning of the Q-landscape. In other words, from a given state, the agent is more likely to have explored more sub-optimal ways to reach the goal. So, when the optimal path is blocked with the barrier, the agents trained with S$^2$AC are more likely to have learned several other ways to go around it. This is different from SAC, when the policy is uni-modal (Gaussian) and the agents are only able to escape the barrier and get to the goal for large $\alpha$'s ($\alpha \in 10, 20$). However, robustness in the case of SAC trained with large $\alpha$'s come at the expense of performance, *i.e.*, increased number of steps (See row 3 in Figure 7). Besides, note that the number of SAC agents reaching the goals for $\alpha = 20$ is less than the one for $\alpha = 10$. This is due to the fact that higher $\alpha$'s lead to higher stochasticity and less structured exploration (the standard

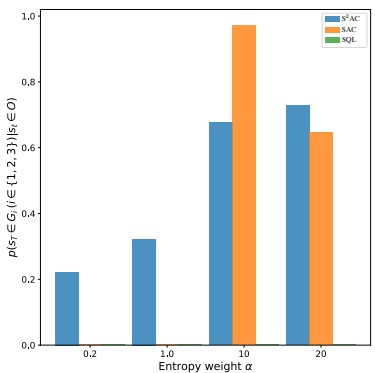

Figure 16: Distribution of reached goals after hitting an obstacle for $S^2$AC, SAC and SQL.

deviation of the Gaussian becomes very large). SQL fails to reach the goals once the obstacle is added. This shows that the implicit entropy in SQL is not as efficient as the explicit entropy in SAC and $S^2$AC.

**Amortized $S^2$AC.** In Figure 17, we report results of the amortized version of $S^2$AC, *i.e.*, $S^2$AC$(\phi, \theta, \psi)$ on the multigoal environment. Performance and robustness are comparable with the non-amortized version $S^2$AC$(\phi, \theta)$ while having a faster inference (feedforward pass through $f_\psi(s, z)$).

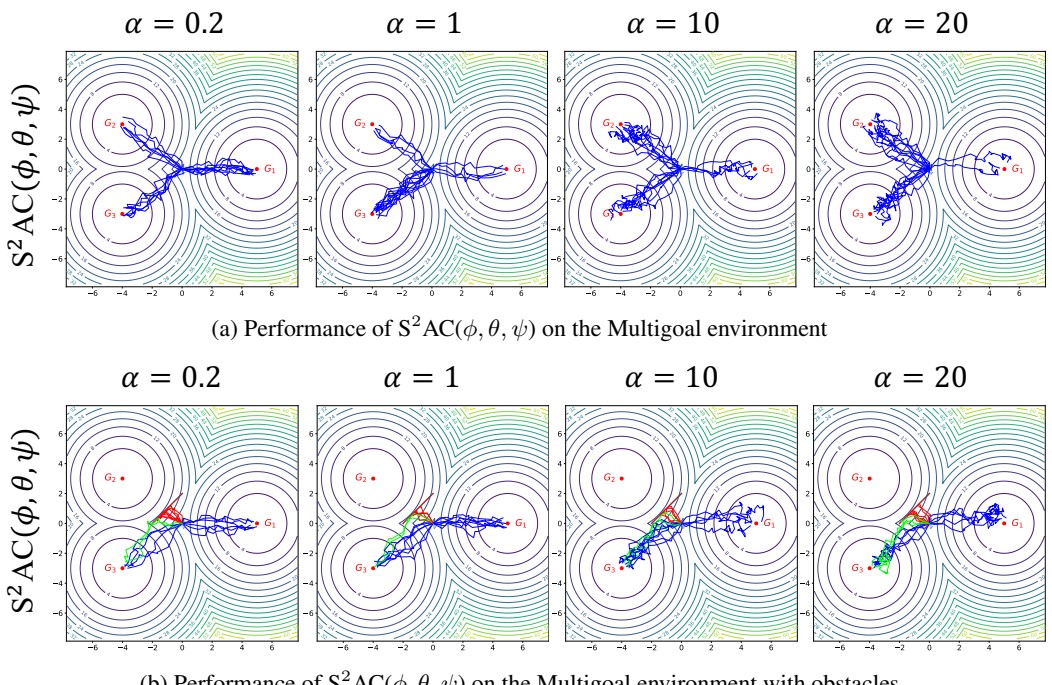

(a) Performance of $S^2$AC$(\phi, \theta, \psi)$ on the Multigoal environment

(b) Performance of $S^2$AC$(\phi, \theta, \psi)$ on the Multigoal environment with obstacles

Figure 17: Performance of Amortized $S^2$AC on the Multigoal environment

## K    ADDITIONAL RESULTS: MUJOCO

Table 4 lists the S$^2$AC hyper-parameters used in our experiments. Additionally, we give details on accelerating S$^2$AC.

Table 4: Hyperparameters

|  | Hyperparameter | Value |
|---|---|---|
| Training | Optimizer
Learning rate
Batch size | Adam
$3 \cdot 10^{-4}$
100 |
| Deepnet | Number of hidden layers (all networks)
Number of hidden units per layer
Number of samples per minibatch
Nonlinearity | 2
256
256
ReLU |
| RL | Target smoothing coefficient
Discount $\gamma$
Target update interval
Entropy weight $\alpha$
Replay buffer size $|\mathcal{D}|$ | 0.005
0.99
1
1.0 for all environments, 0.2 Ant
$10^6$ |
| SVGD | initial distribution $q_0$
Learning rate $\epsilon$
Number of steps $L$ (S$^2$AC($\phi$))
Number of steps $L$ (S$^2$AC($\phi, \theta$))
Number of particles $m$
Particles range (num. std) $t$
Kernel variance | $\mathcal{N}(\mathbf{0}, 0.5\boldsymbol{I})$
0.1
20
3
10
3
$\sigma = \frac{\sum_{i,j} \|a_i - a_j\|^2}{4(2 \log m + 1)}$ |

**Computational Efficiency.** Compared to SAC, running SVGD for $L$ steps requires $L$ additional back-propagation passes through the Q-network and a factor of $m$ (number of particles) increase in the memory complexity. In order to improve the efficiency of S$^2$AC, we limit the number of particles $m$ to 10/20 and the number of SVGD steps $L$ to 10/20.

Additionally, we experiment with the following amortized version of S$^2$AC. Specifically, we train a deepnet $f_\psi(s, z)$ to mimic the SVGD dynamics during testing, where $z$ is a random vector that allows mapping the same state to different particles. Note that we cannot use this deepnet during training as we need to estimate the closed-form entropy which depends on the SVGD dynamics. One way to train $f_\psi(s, z)$ is to run SVGD to convergence and train $f_\psi(s, z)$ to fit SVGD outputs. This however requires collecting a large training set of state action pairs by repeatedly deploying the policy. This might be slow and result in low coverage of the states that are rarely visited by the learned policy and hence result in poor robustness in case of test time perturbations. We instead propose an incremental approach in which $\psi$ is iteratively adjusted so that the network output $a = f_\psi(s, z)$ changes along the Stein variational gradient direction that decreases the KL divergence between the policy and the EBM distribution, *i.e.*,

$$\Delta f_\psi(z, s) = \frac{1}{m} \sum_{i=1}^{m} k(a_i, f_\psi(s, z)) \nabla_{a_i} Q(s, a_i) + \alpha \nabla_{a_i} k(a_i, f_\psi(s, z)) \tag{24}$$

Note that $\Delta f_\psi$ is the optimal direction in the reproducing kernel Hilbert space, and is thus not strictly the gradient of Eq.(5), but it still serves a good approximation, *i.e.*, $\frac{\partial J}{\partial a_t} \propto \Delta f_\psi$, as explained by Wang & Liu (2016). Thus, we can use the chain rule and backpropagate the Stein variational gradient into the policy network according to

$$\frac{\partial J(s)}{\partial \psi} \propto \mathbb{E}_z \left[ \Delta f_\psi(s, z) \frac{\partial f_\psi(z, s)}{\partial \psi} \right]. \tag{25}$$

to learn the optimal sampling network parameters $\psi^*$. Note that the amortized network takes advantage of a Q-value that estimates the expected future entropy which we compute via unrolling the SVGD steps using Eq (3.3).

The modified S$^2$AC algorithm is described in Algorithm 2.

---

**Algorithm 2** Stein Soft Actor Critic (S$^2$AC) with Amortized policy (test-time)

---

1: Initialize parameters $\phi$, $\theta$, $\psi$, hyperparameter $\alpha$, and replay buffer $\mathcal{D} \leftarrow \emptyset$
2: **for** each iteration **do**
3:     **for** each environment step $t$ **do**
4:         Sample action particles $\{a\}$ from $\pi_\theta(\cdot|s_t)$
5:         Select $a_t \in \{a\}$ using exploration strategy
6:         Sample next state $s_{t+1} \sim p(s_{t+1}|s_t, a_t)$
7:         Update replay buffer $\mathcal{D} \leftarrow \mathcal{D} \cup (s_t, a_t, r_t, s_{t+1})$
8:     **for** each gradient step **do**
9:         **Critic update:**
10:           Sample particles $\{a\}$ from an EMB sampler $\pi_\theta(\cdot|s_{t+1})$
11:           Compute entropy $\mathcal{H}(\pi_\theta(\cdot|s_{t+1}))$ using Eq.(11)
12:           Update $\phi$ using Eq.(8)
13:         **Actor update:**
14:           Update $\theta$ using Eq.(9)
15:           Update $\psi$ using Eq.(25)

---

**Evaluation with the Rliable Library.** Performances curves in Figure 8 are averaged over 5 random seeds and then smoothed using Savitzky-Golay filtering with window size 10. Additionally, we report metrics from the Rliable Library (Agarwal et al., 2021) in Fig. 8, including

- **Median**: Confidence interval of the median performance of each algorithm across different seeds, averaged over different MuJoCo environments.
- **Mean**: Confidence interval of the average performance of each algorithm across different seeds and environments.
- **IQM (Interquantile means)**: Instead of computing the average performance on all trials, IQM shows the mean of the middle 50 percent of performance across different seeds.
- **Optimality Gap**: The area between results curve of baseline algorithms and the horizontal line at the average performance of S$^2$AC $(\phi, \theta)$.
- **Probability of improvement over baselines**: The average probability that S$^2$AC $(\phi, \theta)$ can make performance improvements over baseline algorithms.

The parameterized version of S$^2$AC has the best performance among baselines in all the considered metrics. It has a probability of $\sim$65% in outperforming SAC-NF and $\sim$80% in outperforming IAF.