# OpenReview forum: "S$2$AC: Energy-Based Reinforcement Learning with Stein Soft Actor Critic"
_ICLR.cc/2024/Conference — ICLR 2024 poster_

### Official Review · Reviewer_FpMW · 2023-10-30

**Soundness:** 2 fair
**Presentation:** 3 good
**Contribution:** 2 fair
**Rating:** 5
**Confidence:** 5

**Summary:**

This paper proposed a new algorithm for MaxEnt RL that can find a policy with better entropy. The main benefit comes from an explicit evaluation of the entropy term thanks to the closed-form entropy of the based measure and invertiability of the transformation. Experimental results demonstrate the effectiveness of the proposed algorithm on finding maximum entropy policy.

**Strengths:**

* The presentation is easy to follow.
* The demonstration can clearly show the differences between SQL, SAC and the proposed method.
* The computation for the invertibility of SVGD can be of independent interest.

**Weaknesses:**

* I believe that the authors missed an important related work [1]. There are lots of similarities between the proposed method and [1], e.g. using SVGD to approximate the energy-based policy parameterized by the $Q$-function. In fact, if [1] does not amortize the policy, I believe it is nearly identical to the proposed methods.

* I feel there are several modifications that make the proposed method alleviating from the concept of SVGD, e.g. the truncation of the particle update. Furthermore, as there will be some discretization error from the gradient flow of KL on Wasserstein space (which is the motivation of SVGD), I'm thinking if the entropy of the discretized SVGD is a proper estimate of the KL on energy-based policy. I'm also wondering that given the $Q$-function, is it possible to directly estimate the entropy of the energy-based policy, which potentially gets rid of the other issues.

* The experimental results in fact do not beat the baseline a lot.

[1] Liu, Yang, et al. "Stein variational policy gradient." 33rd Conference on Uncertainty in Artificial Intelligence, UAI 2017

**Questions:**

* The relationship between the proposed methods and the references I mentioned in the weakness part.
* Is it possible to directly estimate the entropy of the energy-based policy?

---

> ### Author Response · Authors · 2023-11-22
>
> - **Reference [1]:**
>   Thank you for bringing this to our attention. We actually intended to cite [1] in the SVGD-augmented RL paragraph (last paragraph in related work on page 9) but we ended up citing a different paper by referencing liu2017stein (identical first author, year, and first word of title). Lines 2-4 are actually comparing [1] to our work. Our work is fundamentally different. In [1], the authors learn a multi-modal distribution over the policy parameters ($\theta$) space, while we learn a single policy with multi-modal action ($a$) spaces. We argue that learning a multi-modal distribution over $\theta$ following [1] doesn't necessarily result in learning diverse policies as it's possible to learn the same mapping from states to actions using different parameterizations due to local-minima symmetries in the deepnet loss function. In contrast, learning multi-modal distributions over action spaces results in diverse actions by design. Besides, in real-world setups, we would like to follow the optimal policy and have the possibility to pick a suboptimal action in case of test-time perturbation, without completely switching to a different policy (with a different parametrization $\theta$). We fixed the citation.
>
> - **Particles truncation:**
>   We did not implement the constraint in Eq.9 by truncating the particles to be within 3 std of the mean. Since we use the change of variable formula, we can only apply invertible transformations on the particles (truncation is not invertible). Instead, we sample more particles than we need to select the ones that stay within the range. Since the Q-landscapes become smoother as the training progresses (entropy maximization effect), there are always enough particles to satisfy the constraint. We added this clarification on page 9 under Eq 9.
>
> - **Estimating the entropy of EBMs:**
>   Estimating the entropy of EBMs ($p(x) = \exp{E(x)}/Z$, $E(x) \in \mathbb{R}$ is an energy function) would require evaluating the intractable partition function $Z$ (integration over an infinite number of configurations of $x$). Hence, directly computing the entropy of EBMs is intractable. In Sec B.2 of the appendix, we review previous approaches approximating this quantity and elaborate on how they compare to our method.
>
> - **MuJoCo experiments:**
>   We updated our experiments. Please check the common reply.

---

### Official Review · Reviewer_4Fze · 2023-11-01

**Soundness:** 3 good
**Presentation:** 3 good
**Contribution:** 3 good
**Rating:** 6
**Confidence:** 3

**Summary:**

The paper proposes a new algorithm, named Stein Soft Actor Critic (S$^2$AC), for policy learning with entropy regularization. The goal is to encourage sufficient reward maximization as well as a reasonable coverage of actions, i.e., return a stochastic policy instead of greedy deterministic one. The algorithm is related to existing methods like SAC and SQL algorithms, however, with improved performance demonstrated in empirical results.

**Strengths:**

The paper is mostly well written and easy to follow. The graphical demonstrations help a lot in understanding key concepts. To my knowledge, the proposed algorithm is new, and the empirical results corroborate the improved performance.

I like the idea behind the S$^2$AC algorithm, which is not complicated but to the point.

**Weaknesses:**

1. Technical issues

Equation (2) seems to be inconsistent with the setting in SQL, as there is no discount factor in Equation (2). Is this a typo (as well as in Equation (3))?

Theorem 3.1 is not rigorous. My main concern is around the condition $\epsilon \lVert \nabla h \rVert_{\infty} \ll 1$. First, this is not a precise statement. Second, under this condition, Equation (10) cannot hold with equality (as already displayed in the appendix, there is an approximately equal argument). My suggestion is to either make a precise statement articulating the relationship between the condition and the final assertion to respect the rigor of a theorem, or make it casual by changing it to be a claim with approximate equality. Afterall, this theorem serves as a benign consequence of preserving the invertibility and as a motivation to properly choose $\sigma$.

In Proposition 3.2, it is better to recall the definition of $\sigma$, as there is an ambiguity of $\sigma$ being the variance of Gaussian or kernel function.

2. Claims in empirical results

I am not confident in the claim that "This empirically confirms that SGLD, DLD, and HMC update rules are not invertible" (beginning of page 7) can be obtained from the entropy is not accurately estimated. Can authors provide more context around it? Moreover, why the target distribution is chosen so, as the mean and covariance are not natural (mean $[-0.69, 0.8]$, and variance ...).

I would suggest to avoid the claim of "maximizing the future reward and maximizing the future entropy". The objective in Equation (2) only maximizes the sum of the future reward and future entropy, which is not to maximize both terms.

**Questions:**

Please see the weakness section.

---

> ### Author Response · Authors · 2023-11-22
>
> 1. Technical Issues
>
> **Discount factor in Eq. 2:**
>    This is correct; there should be a discount factor. We added it. We originally followed the notation in the SAC paper which omits the discount factor. We agree that it’s more rigorous this way.
>
>    **Theorem 3.1:**
>    We agree. We changed the equality sign to an approximately equal sign and added the order of the approximation error $\mathcal{O}(\epsilon^2 Ld)$.
>
>    **Definition of $\sigma$:**
>    We updated Proposition 3.2 to include the definition $\sigma$.
>
> 2. Claims in empirical results
>
> **Sec 4.1:**
>    We replaced the sentence "This empirically confirms that SGLD, DLD, and HMC update rules are not invertible" with "This empirically **supports** Proposition 3.4 that SGLD, DLD, and HMC update rules are not invertible".
>
>    **Sec 4.1:**
>    We agree. We replaced "maximizing the future reward and maximizing the future entropy" with "maximizing the sum of the future reward and future entropy" as suggested.

---

> > ### Comment · Reviewer_4Fze · 2023-11-23
> >
> > Thanks for the response. I will keep the score.

---

> > > ### Author Response · Authors · 2023-11-23
> > >
> > > Dear Reviewer 4Fze,
> > >
> > > Thank you once again for your valuable feedback. We have diligently incorporated your suggestions into the manuscript. We are eager to engage in further discussions and address any additional concerns you may have.
> > >
> > > Best regards

---

### Official Review · Reviewer_4hdR · 2023-11-03

**Soundness:** 3 good
**Presentation:** 3 good
**Contribution:** 3 good
**Rating:** 6
**Confidence:** 4

**Summary:**

The authors propose a method that combines soft actor-critic and Stein variational gradient descent to solve the reinforcement learning problem with continuous state and action space. The critic is a common Q-learning method. For the actor, the authors learn an initial distribution for the action that is close to the desired one, in order to improve the sufficiency of the algorithm. The authors provided detailed numerical tests to validate their algorithm.

**Strengths:**

The combination of soft actor-critic and Stein variational gradient descent is a good idea. The presentation of the is clear. The numerical results are comprehensive.

**Weaknesses:**

Certain sections of the article, particularly the basic setting and theorems, are somewhat unclear and could benefit from additional clarification. A more rigorous treatment from the authors in these areas would greatly enhance the overall quality of the paper. Further details can be found in the "Questions" section.

**Questions:**

1.	Page 3. The notation \rho_\pi. Is it the stationary distribution of the state under \pi, or the state distribution s_t, which depends on the initial distribution? If it is the former, please clarify. If it is the latter, then \rho_\pi should depend on t.
2.	Page 3 basic setting. It seems that the authors are considering an RL problem with infinite horizon and without discount. With such a setting, it is not a trivial issue to guarantee the total reward is finite. However, the authors make no comments on this issue.
3.	Page 3 eqn (3). The equivalence between MaxEnt RL and (3) is not clear to me. Can you also provide a short proof in the appendix? It is also surprising to me that (3) does not involve \alpha. (Is there any typo here?)
4.	Page 3 eqn (4). -\alpha should be +\alpha?
5.	Page 4 bottom. In the change of variable formula, there should be an epsilon after I+?
6.	Page 5 equation (9). The motivation for the actor update is to minimize the KL divergence between the policy distribution and EBM of Q-values, which is clear to me. But it is not obvious that minimizing this KL divergence is equivalent to (9). Can you also add a short proof in the appendix. Also, I think \mathcal{D} is used without definition. Q(a|s) looks like a typo, is it Q(a,s) or q(a|s)?
7.	Page 5 theorem 3.1. Eqn (10) should not be equal, but an approximation. Maybe it is better to add the order of the approximation error. I think it is O(\epsilon^2 * L), provided sufficient regularity.
8.	Page 6. Prop 3.5 says HMC is not invertible, but the following paragraph says, “While the HMC update is invertible”. Is it a contradiction?
9.	Related work. I think the paper “Single Timescale Actor-Critic Method to Solve the Linear Quadratic Regulator with Convergence Guarantees” (JMLR 2023) could also be added to the related work. The LQR setting also has continuous state and action space, and the actor is a soft policy.
10.	Page 17. The first \epsilon should not appear in the second last line.

---

> ### Author Response · Authors · 2023-11-22
>
> We really appreciate the reviewer for checking our notations and mathematical soundness in great detail. We are delighted to improve the clarity of our paper based on your feedback.
>
> 1. $\rho_\pi(s_t)$ is the state marginal induced by $\pi$ which does depend on the initial distribution. By convention (e.g., in the Sutton-Barto RL book and the SQL paper), we omitted the transition probability for simplicity.
>
> 2. Thanks. We added a discount factor. We originally followed the notation in the SAC paper which omits the discount factor. We agree that it's more rigorous this way.
>
> 3. We added the proof to Appendix D. On a high level, given the definition of the soft Q-function (Eq. 17), we introduce the Lagrangian of the MaxEnt objective (Eq. 16) by incorporating the constraint $\int_a \pi(a|s)=1$. Computing the derivative w.r.t $\pi(a|s)$ for a given action $a$ and state $s$ results in the exponential policy in Eq. 19. Finally, we choose the Lagrange multiplier such that the obtained distribution sums to one (Eq. 20). We obtain $\pi(a|s)= \exp{( \frac{1}{\alpha} Q(s,a) )}/Z$. $\alpha$ was previously hidden in $Z$. We revised it to reflect the dependency on $\alpha$ in Eq. 3.
>
> 4. That is correct. We revised it.
>
> 5. Thanks. We added it.
>
> 6. We added the proof in Appendix E. We added the definition of $\mathcal{D}$ (replay buffer) and fixed the typo ($Q(s_t,a_t)$ instead of $Q(s_t|a_t)$).
>
> 7. Great point. We changed to using the approximation sign instead and added the order of the approximation error $\mathcal{O}(\epsilon^2 Ld)$. $L$ is the number of SVGD steps and $d$ is the action dimension.
>
> 8. Thanks. We clarified. The HMC update rule is only invertible with respect to the $(a,v)$ (Neal et al. (2011)), i.e., when conditioning on $v$. Since $v$ is sampled from a random distribution, it has the effect of the noise variable in SGLD. Hence, HMC is not invertible w.r.t $a$.
>
> 9. Yes, we added the citation.
>
> 10. Thanks. Corrected.

---

> > ### Comment · Reviewer_4hdR · 2023-11-22
> > **The revision looks good**
> >
> > Thanks the authors for the revision. I suggest that this paper be accepted.

---

> ### Author Response · Authors · 2023-11-23
>
> Dear Reviewer 4hdR,
>
> Thank you for updating your recommendation to "accept". Could you please adjust your score to reflect your recommendation?
> We are eager to engage in further discussions and address any additional concerns you may have.
>
> Best regards

---

### Official Review · Reviewer_4Btw · 2023-11-04

**Soundness:** 3 good
**Presentation:** 3 good
**Contribution:** 3 good
**Rating:** 8
**Confidence:** 3

**Summary:**

This paper argues that in MaxEnt RL, when policy is based on EBMs, estimating the entropy of policies could be an issue. The authors discussed related work, such as SQL and SAC, which have issues to not learn optimal regularized policies. They then proposed S2AC, which learns a more optimal solution to the MaxEnt RL objective. This is achieved by modelling the policy as a Stein Variational Gradient Descent (SVGD) sampler. They show that both SQL and SAC can be recovered with small modifications over S2AC. The authors then conducted experiments to verify the capability of S2AC in estimating entropy, in learning multimodal policies and maximizing entropy, as well as its performance in MuJoCo tasks.

**Strengths:**

1. Estimating entropy in learning MaxEnt RL is a quite interesting question and the authors showed that this actually matters.
2. The idea of using the invertibility of samplers and methods are novel to my knowledge.
3. The experimental results are promising, and they support the proposed methods.
4. The backbone of S2AC, i.e., the new variational inference algorithm, could be of independent interest.
5. The paper is well written. I can easily follow the logic.

**Weaknesses:**

I understand that the paper mostly focuses on discussions with SQL and SAC. However, from reading the paper it seems calculating/estimating entropy of high dimensional policies is an essential part. It could be worth discussing existing literature on entropy estimation and compare the proposed estimation method with them in terms of accurately estimating entropy and computational efficiency.

**Questions:**

1. Does the result apply to other regularizers other than standard entropy?
2. There are also variants of SAC for discrete settings. Do you think the S2AC would benefit in those scenarios?

---

> ### Author Response · Authors · 2023-11-22
>
> We appreciate the reviewer's highly positive comments.
>
> **Literature review of entropy estimation**
>
> In Appendix B, we reviewed (1) a non-parametric entropy estimator that leverages samples from a distribution without access to the normalized density in (Sec B.1) as well as (2) entropy estimators for EBMs (under knowledge of the unnormalized density). In summary, previous techniques are either based on heuristic approximation (e.g., computing pairwise distance between samples as a proxy for smoothness), lower bounds, or neural estimators of mutual information.
>
> Differently, our approach is: (1) more principled as it computes the entropy of the actor (as opposed to using off-the-shelf generic tricks) by exploiting the invertibility of the SVGD update and the change of variable formula, (2) more parameter-efficient (we only learn a Gaussian initial distribution).
>
> **Generalizing $S^2AC$ to discrete settings**
>
> Generalizing $S^2AC$ to discrete settings is non-trivial as SVGD is by default a sampler for continuous probabilities. Since the actions in discrete setups are **finite** and pre-defined, the policy is a softmax distribution with dimensions corresponding to the number of actions. Hence, the entropy can be obtained by computing the empirical mean of the negative log-likelihood of this distribution over all actions.

---

### Official Review · Reviewer_ThdK · 2023-11-04

**Soundness:** 3 good
**Presentation:** 3 good
**Contribution:** 2 fair
**Rating:** 6
**Confidence:** 3

**Summary:**

The author present an algorithm to use steins gradient descent to learn stochastic policies using soft-actor critic. The specific advantage is to not rely on variational approaches  (such as Gaussian-like) for the stochasticity of the policy.

**Strengths:**

- The paper is well written
- Methodology appears sound and novel
- Related literature is checked and carefully introduced  (e.g. Relations to SQL and SAC)
- Design choices (such as Parameterized initialization and amortized inference) are valid and thoroughly discussed
- Theoretical insights, such as a closed form expression of the entropy and invertibility of SVGD  are given

**Weaknesses:**

Significance:  My biggest reservation for acceptance is significance.  How much does the increased expressiveness of the stochastic policy  (at the cost of computation) matter? The authors could only show significant differences to existing methods on hand-crafted toy problems. For the traditional benchmarks, i.e. Mujoco the method does not appear to perform better than standard approaches such as PPO. The problem here, is not the fact that it not performs better, but that it appears better experiments could be designed to show how large/if there is a significant advantage.

The general arguments for stochastic policies are, to my knowledge e.g.  "multi-modality also has application in real robot tasks, as demonstrated in (Daniel et al., 2012) Quote taken from SAC paper. Additionally, as the authors write themselves: "this enables better explo-
ration during training and eventually better robustness to environmental perturbations at test time, i.e., the agent learns multimodal action space distributions which enables picking the next best action in case a perturbation prevents the execution of the optimal one."

If that is the argumentation I would expect the authors to design experiments under these scenarios that are not toy.  This could have been achieved for instance by  augmenting the MuJoCo tasks with perturbation events that prevent execution of certain actions in certain situations. Then a more expressive stochastic policy could perform better.  For now it makes it hard to asses if the increase time- and algorithmic complexity is worth the cost of obtaining better performing policies.

**Questions:**

- How does the armotized version of the method perform on all benchmarks?  Figure 9 (right) only shows one performance curve with no indication which benchmark it is
- Because the armotized version is much faster and performs similarly I wonder why it is not the main method, but it details are hidden in the appendix.
- Also I would be curious how this version performs on the toy problems  (Fig. 5-7).

---

> ### Author Response · Authors · 2023-11-22
>
> **Amortized $S^2AC$**
>
> Results are added to Fig. 8 ($S^2AC(\phi,\theta,\psi)$). There is a small tradeoff in performance for 3 times faster inference compared to the non-amortized version ($S^2AC(\phi,\theta)$). Yet, the improvement over the baselines is still statistically significant (Fig.8 f-j). We added more details on the amortized version in Sec. 3.4. Note that the amortization is just a trick to improve the **test-time**. Specifically, amortized S2AC is just training a feedforward network $f_{\psi}(s,z)$ (Eq.24-25) to mimic the rolled-out (non-amortized policy) as test time (L.15, Alg2). **During training**, we need to use the unrolled policy to be able to compute the entropy estimate of the sampler in the MaxEnt RL objective (L.11-12 in Alg2). Depending on the application, it can be used as the main method, at test time, i.e., if timely reactivity is important (e.g., robot manipulation tasks). In other setups where the accuracy matters more (e.g., medical diagnosis), the unrolled policy would enable checking the Stein identity to assess convergence.
>
> **Results of Amortized $S^2AC$ on multi-goal** are reported in Fig.17.
>
> **Significance on complex environment**: Humanoid-v2 (Fig. 8-e) is a multi-modal environment (the action space is 17 dimensional with a large number of continuous control factors including combinations of legs and arms movements, posture, $\cdots$). Note that our approach (violet) is more sample efficient and performs better than SAC (red) due to the improved ability to explore different local optima. We thought about testing robustness by modifying the humanoid environment, for instance by freezing some variables in the action space to force the robot into a different forward movement mode. This is, however, non-trivial as it impacts the robot's balance. We agree, as RL algorithms are being deployed in real-world setups, ensuring the robustness of these agents is becoming increasingly important. We added a sentence in the conclusion to encourage the community to invest more in this direction.

---

### Official Review · Reviewer_1y8M · 2023-11-06

**Soundness:** 4 excellent
**Presentation:** 4 excellent
**Contribution:** 3 good
**Rating:** 6
**Confidence:** 3

**Summary:**

This paper proposes a novel maximum entropy reinforcement learning algorithm that leverages Stein Variational Gradient Descent (SVGD) that allows the analytic computation of entropy.

**Strengths:**

* The exposition of the paper is clear. The paper explicitly presents its problem, and the proposed solution explicitly addresses the problem.
* The proposed method is theoretically sound and enjoys interesting connections to existing algorithms.
* The paper's claims are well supported by the toy experiments. Experiment results on Mujoco environment seem promising.

**Weaknesses:**

* The main limitation seems to be the slow inference of SVGD. However, the paper argues that this limitation can be addressed by amortization. It would be better if the main manuscript provided more details on how amortization is performed.
* It would be great if the paper mentions the scalability of SVGD with respect to dimensionality. How large the method can be scaled in terms of the dimensionality of the problem? Scaling SVGD to higher dimensional spaces seems to be challenging because we need exponentially more particles to represent a distribution.

**Questions:**

See weaknesses.

---

> ### Author Response · Authors · 2023-11-22
>
> **[Amortized S$^2$AC]**
>
> We added more details in Sec 4.3 (run-time). The full description is as follows. Unfortunately, we cannot include all the details in the main manuscript because of the strict page limit, and it has been a tough decision to put them into the appendix.
>
> Basically, we train a deepnet $f_{\psi}(s,z)$ to mimic the SVGD dynamics during testing, where $z$ is a random vector that allows mapping the same state to different particles. One way to train $f_{\psi}(s,z)$ is to run SVGD to convergence and train $f_{\psi}(s,z)$ to fit SVGD outputs. This, however, requires collecting a large training set of state action pairs by repeatedly deploying the policy. This might be slow and result in low coverage of the states that are rarely visited by the learned policy and hence lead to poor robustness in case of test time perturbations. We instead propose an incremental approach in which $\psi$ is iteratively adjusted so that the network output $a = f_{\psi}(s,z)$ changes along the Stein variational gradient direction that decreases the KL divergence between the policy and the EBM distribution, i.e.,
>
> $$
> \Delta f_{\psi}(z,s) = \frac{1}{m} \sum_{i=1}^{m} k(a_{i}, f_{\psi}(s,z)) \nabla_{a_i} Q(s,a_i) + \alpha \nabla_{a_i}k(a_i,f_{\psi}(s,z))
> $$
>
> Note that $\Delta f_{\psi}$ is the optimal direction in the reproducing kernel Hilbert space, and is thus not strictly the gradient of Eq.(5), but it still serves a good approximation, i.e., $\frac{\partial J}{\partial a_t} \propto \Delta f_{\psi}$, as explained by Wang & Liu (2016). Thus, we can use the chain rule and backpropagate the Stein variational gradient into the policy network according to
>
> $$
> \frac{\partial J(s)}{\partial \psi} \propto E_{z} \left[ \Delta f_{\psi}(s,z)  \frac{\partial f_{\psi}(z,s)}{\partial \psi}  \right].
> $$
>
> to learn the optimal sampling network parameters $\psi^{*}$. Note that the amortized network takes advantage of a Q-value that estimates the expected future entropy which we compute via unrolling the SVGD steps. The modified \STAC\ algorithm is described in Algorithm 2.
>
> **[Scalability of SVGD with respect to dimensionality]**
>
> Theoretically, SVGD does suffer from the curse of dimensionality for arbitrary distributions. But empirically, S$^2$AC($\phi$, $\theta$) works well on the MuJoCo benchmarks with high-dimension action spaces, e.g., the Humanoid environment with a 17-dimensional action space. This may depend on the specific shape of the distribution, and the fact that we have a **parameterized initialization** of the initial particle distribution has greatly mitigated the issue since it learns to focus on the sub-spaces for action/particle sampling. Notice the difference in performance between the parametrized S$^2$AC($\phi$, $\theta$) and non-parametrized S$^2$AC($\phi$) versions in Fig. 8. In the future, we aim to further improve the scalability by stacking multiple SVGD-induced distributions in a diffusion model style.

---

### Official Review · Reviewer_mSs2 · 2023-11-06

**Soundness:** 3 good
**Presentation:** 2 fair
**Contribution:** 2 fair
**Rating:** 3
**Confidence:** 4

**Summary:**

This paper deals with creating an SAC like algorithm, but while allowing
multimodality of the policy. This is achieved by using a Stein Variational
Gradient Descent sampler. A neural network parameterizes the mean and variance
of the initial distribution of the sampler. Moreover, the samples produced
by the SVGD method are restricted to lie within 3 standard deviations of the
initial distribution. The policies entropy can be computed thanks to the
SVGD sampling process being invertible, similar to how a normalizing flow
works.

There are experiments to check the correctness of the formulation in simple 2D landscape fitting tasks. Moreover, they perform experiments
on 5 MuJoCo benchmark tasks. The performance seems similar to SAC in
4/5 tasks, and better in one of the tasks.

**Update**
____________________________

Thanks for the extensive update.

Many things improved, but I also still have concerns.

Mainly, the rebuttal claims: "SSPG ... Specifically, the middle plot in Fig. 4(A) is an identical setup to the Multi-goal environment in Fig. 2 in our paper." But this does not seem to be true. The task in the SSPG is a bandit problem with only 1 environment step (and the action selects the location on the landscape); hence, there is no "future entropy" as there is just one step in the environment. Whereas in the current paper, there is an agent that moves around on the landscape, so the tasks are different. I see no reason why SSPG would also not be able to maximize the future entropy, as it uses a soft critic that includes the future entropy similarly to this paper.

Another concern is that your re-implementation of SAC-NF does not improve over SAC, whereas it did in the references, both in the SAC and REDQ cases, so perhaps your implementation is not well-performing.

While the use of the rliable library was a good step to improve the analysis, there is a fairly large overlap in the error bars for SAC and your newly proposed method. As the number of random number seed experiments per environment was 5, I think this should be increased to reduce the error bars and receive a more reliable result.

Based on the above, I decided to keep my score. What may have changed my assessment would have been performing more experiments so that the statistical significance of the results is clear (this would have increased my score to 5), not making claims that the setup is the same as the one in SSPG, better results for SAC-NF, etc. However, I will increase the contribution rating by 1 point, as I think there are also other contributions in the work.

[3] SSPG: "Policy Gradient With Serial Markov Chain Reasoning"

**Strengths:**

The correctness of the method is properly checked with experiments.

**Weaknesses:**

- There are many works looking at multimodality in MaxEnt RL that were
not discussed or cited. Moreover several of these obtain better
results than the current work (although they include additional features).
For example one earlier work is "Boosting trust region policy optimization with
normalizing flows policy" (https://arxiv.org/pdf/1809.10326.pdf). Their equation
(5) seems similar to equations (10) and (11) in the current paper (it's more
clear from the proof sketch). Moreover, there are the recent works:
"Reparameterized Policy Learning for Multimodal Trajectory Optimization"
(https://arxiv.org/pdf/2307.10710.pdf), although it is model-based, and
also the work: "Policy Gradient With Serial Markov Chain Reasoning"
(https://openreview.net/forum?id=5VHK0q6Oo4M) that is model-free and obtains
good performance (it's also based on the MaxEnt principle); this work
also included a comparison with a normalizing flow in their appendix,
and showed that their method achieves better performance; the experiments
in this work were also more substantial than the current work.
Finally, there are some other works that can be found in the related
works section of this paper: "Leveraging exploration in off-policy algorithms via
normalizing flows" (https://arxiv.org/pdf/1905.06893.pdf),
"Iterative Amortized Policy Optimization" (https://arxiv.org/pdf/2010.10670.pdf).
None of these works were cited, so I think the literature review was insufficient
(they cite many works, but miss the most relevant ones).

- The experimental results are not very strong. The improvement over
SAC is marginal. The other works I referenced above have more substantial experiments,
and show greater improvements.

**Questions:**

How does your method compare to other methods that create multi-modal
policies in RL? Please include a comparison.

How is it related to normalizing flows?

---

> ### Author Response · Authors · 2023-11-22
> **Clarification of suggested reference and new experiments**
>
> Thank you for sharing the references. We tried to conduct an extensive literature review, but these references were missed because their titles/abstracts do not indicate a strong relation to our work.
>
> We carefully studied them.
>
> **TL;DR: MaxEnt RL is not just about multimodality**. First, we want to emphasize that our goal is to find a better solution to the MaxEnt RL objective due to its nice properties. Notably, beyond better exploration and multi-modality, MaxEnt RL is provably *robust*. This is achieved by maximizing the future entropy (positive reward setup) and is a highly desired property in real-world applications. Our empirical experiments (Fig. 6) validate that $S^2AC$ is not only multi-modal but also robust. This is, however, not the case for Ref [1-5]. Next, we argue that **Refs [1,2]** solve different objectives that lead to multi-modal policies but do not optimize for robustness. While **Refs [3,4,5]** optimize the MaxEnt RL objective, the learned policies fail to maximize the expected future entropy because of approximations (**Ref [3]**), the instability of training flow models (**Ref [4]**), or modeling the policy as uni-modal Gaussian (**Ref [5]**). We included **a comparison to these methods** in the intro and related work sections. We also **re-implemented Ref [4]** for comparison in both the Multi-goal and MuJoCo experiments (their code is not available).
>
> Detailed clarifications are as follows.
>
> **Refs [1,2] optimize different objectives from MaxEnt RL.** **Ref [1]** optimizes Trusted Region Policy Optimization with a KL-divergence constraint. **Ref [2]** solves a new ELBO (Eq2) which is inspired but different from the MaxEnt RL objective. Specifically, it introduces additional latent variables and follows a model-based formulation.
>
> **Refs [3,4,5] fail to maximize the future entropy.**
> - **SSPG (Ref [3])** models the policy as a Markov chain with transition probabilities being parametrized Gaussians, which enables computing an approximation of the policy gradient (Eq. 13). The experimental results in Figure 4(A) don't demonstrate maximizing the future entropy. Specifically, *the middle plot in Fig. 4(A)* is an identical setup to the Multi-goal environment in Fig. 2 in our paper. As stated by the authors (lines 9-10 of Sec. 4.2), SSPG learns to visit the goals with *similar frequencies*. However, a good solution to the MaxEnt RL objective would be biased towards sampling from the lower modes. Hence, it seems that SSPG has just led to better exploration without robustness benefits. We wanted to further investigate, but the author didn't provide the code (empty link). In contrast, we show that explicitly computing the entropy leads to more optimal MaxEnt RL solutions (Fig. 2), *i.e., to maximizing the expected future entropy*.
> - **SAC-NF (Ref [4])** models the policy as a normalizing flow (NF). **We re-implemented the paper** (their code was not provided) and reported results on both the Multi-goal (**Figs. 6-7**) and MuJoCo envs (**Fig. 8**). On Multi-goal, SAC-NF learns to maximize the entropy but, unlike our approach, it misses one mode for smaller $\alpha$ values. We argue that this is due to normalizing flow well-known tendency to quickly collapse to local optima Kobyzev et al. (2020) (in accordance with Fig. 4(A) in Ref[3]). The dispersion term in S2AC encodes an inductive bias to mitigate this issue. On MujoCo, convergence is slower and performance is lower than our approach. **Note that the reported results in Ref [3] are from applying NFs over REDQ (Table 5)**.
> - **IAP (Ref [5])** models the policy as a uni-modal Gaussian (Sec. 4.1). Multi-modality is achieved by learning a collection of parameter estimates (mean, variance) through different initializations (Figure 2 and Sec. 3.2). The estimates result in parametrizing different policies. This doesn't result in maximizing the entropy and hence doesn't lead to robustness.
>
> **Eq 5 in Ref [1]** is the Change of Variable Formula (CVF), which we use as a tool given that SVGD is invertible (similarly to normalizing flows). Our theoretical contribution consists of (1) proving that SVGD is invertible which enables the use of CVF, (2) using CVF together with the corollary of Jacobi’s formula to derive a closed-form expression of the SVGD log-likelihood (Eq 10), (3) deriving an equivalent computationally efficient expression of Eq. 10 (evaluates the trace of the Jacobian) that only depends on first order derivatives and vector dot products (Eq. 11) and (4) optimizing for scalability via parameterizing Eq. 11.
>
> **Experiments.** We updated our experiments. Please check the common reply.
>
> - [1] Tang, Yunhao et al. ICLR19
>
> - [2] Huang, Zhiao, et al. 2023.
>
> - [3] Cetin, Edoardo, et al. NeurIPS, 2022.
>
> - [4] Mazoure, Bogdan, et al. CoRL, 2020.
>
> - [5] Marino, Joseph, et al. NeurIPS, 2021.

---

### Author Response · Authors · 2023-11-22
**Summary of key aspects and updated experiments**

We thank the reviewers for the valuable feedback and the positive comments on the strengths of our paper:

- **The technical novelty in deriving a closed-form expression of the distribution induced by SVGD dynamics.** Our formula is a new variational inference distribution of independent interest (beyond RL) and compelling properties, as it is (a) highly expressive, (b) easy to sample from (SVGD is more particle efficient than MCMC), (c) parameter efficient (same number of trainable parameters as SAC), (d) computationally efficient (only depends on vector dot products and first-order derivatives), (e) scalable (via parametrizing the initial distribution and constraining the particles update) and (f) with convergence to the target distribution easily assessed via the Stein-Identity.

- **The experimental design supporting the theory.** We evaluated (1) the correctness of the entropy estimate on a distribution with a tractable GT entropy (Sec 4.1), (2) convergence to a more optimal solution of the MaxEntr RL objective in the Multi-goal environment (Sec 4.2) and (3) performance on the MuJoCo benchmark (Sec 4.3).

- **Clarity of the presentation.**

**[Updated experiments].**

1. **Multi-goal:**
    - We added results for (1) SAC-NF (SAC with a normalizing flow policy) in Figs. 6-7.
    - We added results for (2) amortized $S^2AC$ in Fig. 17.

2. **MuJoCo:**
    - We included the flow model as an additional baseline (Fig. 8).
    - We reported results for amortized $S^2AC$, i.e., $S^2AC(\phi, \theta, \psi)$ for all the environments (Fig. 8).
    - For a more comprehensive comparison against baselines, we reported additional metrics from the Reliable library [1] in Fig. 8, including normalized performance profiles, Interquantile means (IQM) and probability of improvement over baselines. The reported ranges/shaded regions represent 95\% stratified bootstrap confidence intervals.
    - For each experiment, we increased the number of evaluations per seed to 100 as opposed to 10 at the time of the submission, and used a smoothing window of size 10 following best practices.

We updated the paper based on the reviewers' feedback. All major changes are highlighted in purple.

[1] Agarwal, Rishabh, et al. "Deep reinforcement learning at the edge of the statistical precipice." NeurIPS, 2021.

---

### Meta-Review · Area_Chair_Y9zW · 2023-12-05

**Metareview:**

This work contributes an interesting usage of Stein Variational Gradient Descent in an actor-critic setting for solving Markov Decision Processes, with substantial experimental analysis demonstrating the merits of the proposed approach. While there is some disagreement over the extent of experimental validation amongst the reviewers, there is enough agreement regarding the merits in theory and practice to warrant acceptance.

In particular, while reviewer mSs2 has expressed some valid criticism of limitations of the experiments, the authors have expended extensive effort to create implementations of competitors during the rebuttal period, which together with the credible theoretical contributions of the work, were enough to move the needle.

**Justification For Why Not Higher Score:**

NA

**Justification For Why Not Lower Score:**

NA

---

### Decision · Program_Chairs · 2024-01-16

Accept (poster)